# Proteo-R1: Reasoning Foundation Models for *De Novo* Protein Design

Fang Wu [* 1]  Weihao Xuan [* 2 3]  Heli Qi [* 3]  Hanqun Cao [* 4]  Heng-Jui Chang [* 1]  Zeqi Zhou [* 2]
Li Erran Li [* 5]  Haokai Zhao [6]  Ma Jian [7]  Carl Ma [1]  Yu-Chi Cheng [8]  Kuan Pang [1]  Xiangru Tang [9]
Zehong Wang [10]  Guanlue Li [11]  Hanchen Wang [1]  Kejun Ying [1]  Pan Lu [1]  Chiho Im [1]  Seungju Han [1]
Peng Xia [12]  Tinson Xu [13]  Yinxi Li [14]  Deyao Zhu [15]  Pheng-Ann Heng [4]  Naoto Yokoya [2 3]  Masashi Sugiyama [2 3]
Jure Leskovec [1]  Yejin Choi [1]

## Abstract

Deep learning in *de novo* protein design has achieved atomic-level fidelity. However, existing models remain largely non-deliberative: they directly synthesize molecular geometries without explicitly reasoning about which residues or interactions are functionally essential. As a result, design decisions are entangled with continuous sampling dynamics, limiting interpretability, controllability, and systematic reuse of biochemical knowledge. We introduce **Proteo-R1**, a reasoning-guided protein design framework that explicitly decouples *molecular understanding* from *geometric generation*. Proteo-R1 adopts a dual-expert architecture, in which a multimodal large language model (LLM) serves as an *understanding expert* and analyzes protein sequences, structures, and textual context to identify key functional residues that govern binding and specificity. These residue-level decisions are then passed to a separate diffusion-based *generation expert*, which performs conditional co-design while respecting the fixed interaction anchors. This factorization mirrors how human experts approach molecular engineering: first, reasoning about critical interactions, then optimizing geometry subject to those constraints. By operationalizing reasoning as explicit residue-level commitments rather than latent textual guidance, Proteo-R1 achieves

stable, interpretable, and modular integration of LLM reasoning with advanced geometric generative models. Code and demos are at `https://smiles724.github.io/r1/`.

## 1. Introduction

Deep generative models based on diffusion and flow matching (Song et al., 2020; Lipman et al., 2023; Yang et al., 2023) have reshaped the landscape of molecular design, enabling *de novo* generation of proteins (Watson et al., 2023), peptides (Li et al., 2024b; Lin et al., 2024c), antibodies (Kong et al., 2025), and small-molecule (Oestreich et al., 2025; Li et al., 2026b) binders with unprecedented structural fidelity. By learning to reverse stochastic dynamics in high-dimensional continuous spaces, these models can directly synthesize atomic coordinates and amino-acid identities conditioned on complex biochemical contexts, dramatically accelerating discovery pipelines that were once dominated by human intuition and iterative experimentation (Alakhdar et al., 2024; Schneuing et al., 2024; Zeni et al., 2025).

However, despite their expressive power, current generative models remain fundamentally *non-deliberative*. They produce molecular structures without explicitly reasoning about which residues are functionally critical, why certain interactions are favored, or how design constraints should be prioritized. In most frameworks, all residues are treated uniformly during generation, and design intent is implicitly encoded in the parameters of the diffusion process. This entanglement of reasoning and generation obscures interpretability, complicates control, and limits the ability to reuse or refine design logic across tasks.

In contrast, human molecular designers rarely operate in this manner. A structural biologist or protein engineer typically begins by identifying key interaction residues (De-Lano, 2002; Wells & McClendon, 2007) – such as charged anchors forming salt bridges, hydrophobic hotspots stabilizing interfaces, or specificity-determining motifs – before optimizing the remaining degrees of freedom. Geometry

---

[*]Equal contribution  [1]Stanford University  [2]University of Tokyo  [3]RIKEN AIP  [4]Chinese University of Hong Kong  [5]AWS AI, Amazon (work outside of Amazon)  [6]University of New South Wales  [7]Shanghai Jiao Tong University  [8]Harvard University  [9]Yale University  [10]University of Notre Dame  [11]University of Hamburg, Germany  [12]UNC Chapel Hill  [13]University of Chicago, USA  [14]University of Waterloo  [15]ByteDance Seed. Correspondence to: Fang Wu <fangwu97@stanford.edu>, Jure Leskovec <jure@cs.stanford.edu>, Yejin Choi <yejinc@stanford.edu>.

*Proceedings of the $43^{rd}$ International Conference on Machine Learning*, Seoul, South Korea. PMLR 306, 2026. Copyright 2026 by the author(s).

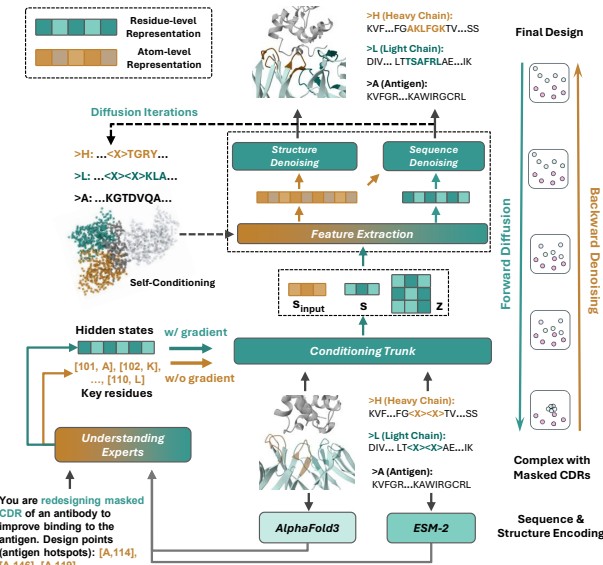

*Figure 1.* Proteo-R1 couples a multimodal reasoning expert with a geometric diffusion expert to unify molecular understanding and generation. The reasoner integrates sequence embeddings, AF3-style structural representations, and text prompts to analyze a masked complex and determine which CDR residues should be key interaction anchors. These decisions include both the selection of critical residues and their preferred biochemical identities or roles. The residue-level constraints are then passed to the generator, which performs conditional co-design throughout the generative process.

is refined *after* high-level decisions are made, not simultaneously with them (Kuhlman et al., 2003; Leaver-Fay et al., 2011b). This separation between *reasoning about what matters* and *optimizing how it is realized* is central to expert-driven molecular design, yet is largely absent from contemporary generative models (Tang et al., 2024).

Motivated by this gap, we propose **Proteo-R1** (Fig. 1), a new paradigm for reasoning-guided protein design that explicitly factorizes molecular understanding from geometric generation. It is a dual-expert framework in which a multimodal large language model (MLLM) serves as an *understanding expert*, while an AlphaFold3 (AF3)-like diffusion model (Stark et al., 2025; Team et al., 2025; Zambaldi et al., 2024) serves as a *generation expert*. Rather than embedding chain-of-thought (CoT) representations directly into a continuous denoising trajectory (Deng et al., 2025), Proteo-R1 operationalizes reasoning through explicit, residue-level decisions that are subsequently enforced during generation.

This yields several important advantages. First, it provides a clear, interpretable interface between reasoning and generation, enabling the inspection, modification, and reuse of design decisions independently of the diffusion model. Second, it allows the explicit incorporation of human prior knowledge by pretraining on large-scale scientific corpora (e.g., PubMed and related biomedical literature). Third, it preserves the stability and inductive biases of state-of-

the-art geometric generators by avoiding direct injection of textual or symbolic representations into continuous dynamics. Last, it enables modularity: the same reasoning expert can guide a wide range of generative backends, including AF3-like design models (Zambaldi et al., 2024), flow-matching architectures (Yu et al., 2026), and other emerging frameworks (Pacesa et al., 2024; Mille-Fragoso et al., 2025).

We evaluate Proteo-R1 in the context of antibody complementarity-determining region (CDR) co-design. Experiments show that explicitly reasoning about key residues before generation yields improved structural realism, binding rationality, and controllability relative to purely generative baselines. More broadly, Proteo-R1 suggests a scalable blueprint for integrating LLMs with physical generative processes: LLMs act not as noisy conditioners, but as molecular strategists that guide design through explicit, biologically grounded decisions. A discussion of related work is provided in Appx. A.

## 2. Method

Proteo-R1 is a *dual-expert* framework that explicitly factorizes *multimodal molecular understanding* from *structure-sequence generation*. It comprises two specialized components: (i) a **multimodal understanding expert** $E_{und}$, which produces residue-level representations encoding biochemical, structural, and contextual information, and (ii) a **generation expert** $E_{gen}$, instantiated as an AF3-style diffusion model (Yang et al., 2026) to perform conditional co-design.

A key design principle of Proteo-R1 is that information flows between the two experts through *explicit, residue-aligned embedding injection*. Hidden representations produced by the understanding expert are projected into the diffusion model's residue embedding space and used to selectively replace the standard <X> embeddings at key CDR positions. This interface enables deterministic incorporation of residue-level reasoning while preserving the inductive biases and stability of the underlying diffusion generator. We train Proteo-R1 using a three-stage curriculum (Fig. 2) that progressively stabilizes cross-modal grounding, geometric reasoning, and end-to-end design.

### 2.1. Preliminaries and Problem Setup

We consider *co-designing CDR sequence and structure* conditioned on an antibody-antigen co-crystal complex. Following Yang et al. (2026); Kong et al. (2025), the antigen sequence and structure, the antibody framework regions (FRs), and the overall antibody-antigen docking pose are assumed to be known and fixed, while CDR sequences and structures are treated as design variables. To formulate this problem as

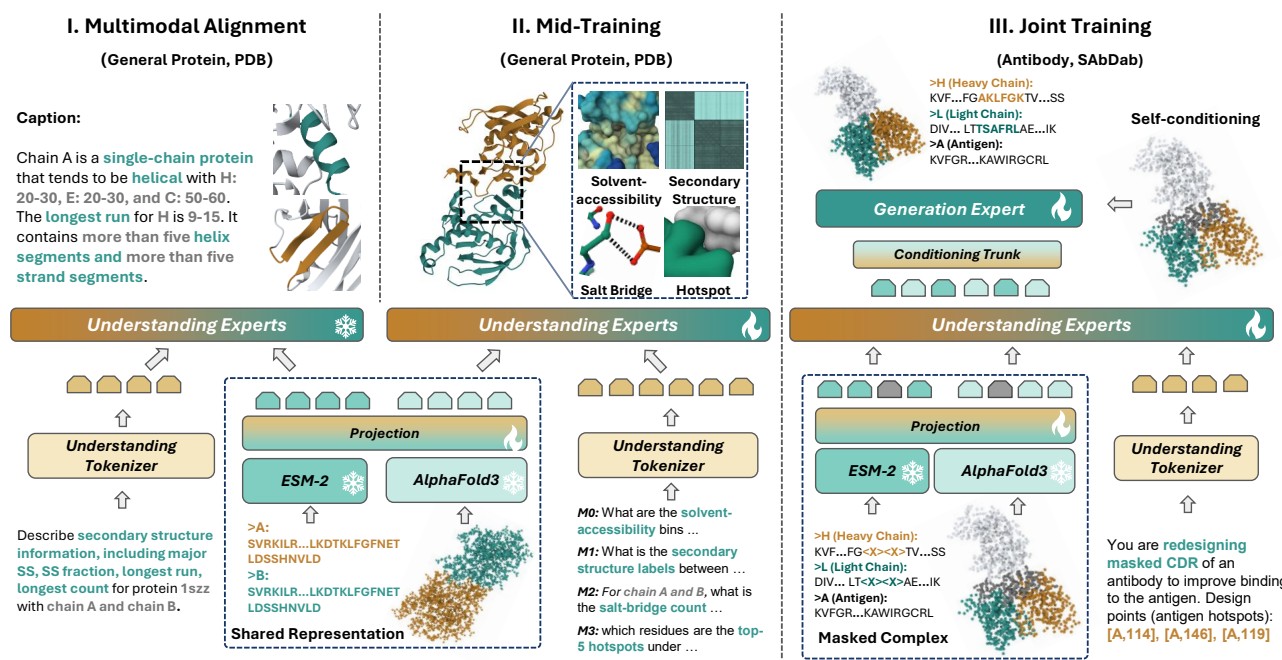

*Figure 2.* **Three-stage training diagram of Proteo-R1.** In Stage I (Multimodal Alignment), the framework uses general protein data from PDB to project sequence and structural features into the LLM's language representation space via lightweight projection layers, while the LLM backbone remains frozen. Supervision combines structured schema completion and free-form captioning over chain-level structural attributes. Stage II (Structural Reasoning Mid-Training) unfreezes the LLM backbone to master a four-phase curriculum of spatial meta-tasks—from residue grounding and pairwise geometry to interface localization and hotspot prediction—equipping $\mathbf{E}_{\text{und}}$ with essential geometric reasoning skills. The last Stage III (Joint Reasoning-Guided Design) performs end-to-end optimization on antibody-antigen complexes from SAbDab, where residue-level reasoning directly steers the diffusion-based CDR redesign process.

conditional generation, all CDR residues are masked at the sequence level by replacing their amino-acid identities with a special unknown token `<X>`. In contrast, residues outside the CDRs remain fully specified. Let the full antibody-antigen complex consist of $N$ residues with atomic coordinates $\mathbf{X} \in \mathbb{R}^{M \times 3}$, and let $\mathcal{I}_{\text{CDR}} \subset \{1, \ldots, N\}$ denote the index set of CDR residues. For each residue $i$, we denote its amino-acid identity by $k_i \in \{1, \ldots, 20\}$ and its full-atom coordinates by $\mathbf{x}_i \in \mathbb{R}^{n_i \times 3}$. The objective of antibody co-design is to learn the conditional distribution

$$p\big(\{k_i, \mathbf{x}_i\}_{i \in \mathcal{I}_{\text{CDR}}} \,\big|\, \{k_j, \mathbf{x}_j\}_{j \notin \mathcal{I}_{\text{CDR}}}, \, \mathcal{C}\big), \tag{1}$$

where $\mathcal{C}$ denotes optional conditioning such as epitope annotations or functional design constraints. Under this formulation, the model replaces `<X>` tokens by jointly generating amino-acid identities and full-atom structures for all CDR residues, yielding a complete antibody consistent with the fixed context.

### 2.2. Multimodal Understanding Expert

$\mathbf{E}_{\text{und}}$ performs *pre-generative molecular reasoning*. Rather than directly participating in geometric synthesis, it analyzes the antibody-antigen complex. It produces residue-level representations that summarize biochemical context, spatial organization, and interfacial relationships, without committing to specific atomic coordinates. These represen-

tations serve as the sole interface through which molecular understanding is passed to the generative diffusion model. Throughout Proteo-R1, $\mathbf{E}_{\text{und}}$ never observes ground-truth CDR structures; all structural features are derived from re-folded predictions conditioned on masked CDR sequences.

**Sequence Encoding.** Given the full antibody-antigen sequence with all CDR residues masked using a special `<X>` token, we obtain contextualized sequence representations using a pretrained protein language model (ESM-2) (Lin et al., 2023). For each residue $i$, the encoder produces $\mathbf{h}_i^{\text{seq}} = f_{\text{ESM}}\big(\{\tilde{k}_j\}_{j=1}^N\big) \in \mathbb{R}^{d_{\text{seq}}}$, which captures evolutionary constraints, biochemical preferences, and long-range sequence context under the masked CDR setting.

**Structure Encoding via CDR-Masked Refolding.** To provide $\mathbf{E}_{\text{und}}$ with structural and inter-chain context while preventing leakage of native CDR geometry, we encode structure using an AF3-style folding model (Abramson et al., 2024; Gong et al., 2025) under a CDR-masked *inpainting* scheme. This formulation follows the replacement sampling paradigm originally developed for image inpainting (Lugmayr et al., 2022) and subsequently adapted to molecular and protein generation, conditioning sampling on fixed motifs while redesigning masked regions (Trippe et al., 2022; Schneuing et al., 2024).

Given an antibody-antigen complex with experimental atomic coordinates $\mathbf{X}^\star \in \mathbb{R}^{M \times 3}$ and residue identities $\{k_j\}_{j=1}^N$, we mask all CDR residues at the sequence level by defining $\tilde{k}_j = \text{<X>}\,\mathbf{1}[j \in \mathcal{I}_{\text{CDR}}] + k_j\,\mathbf{1}[j \notin \mathcal{I}_{\text{CDR}}]$. Conditioned on the unmasked framework and antigen residues, the masked complex is then refolded using an AF3-style model, yielding imputed coordinates $\tilde{\mathbf{X}}$. This process treats CDRs as missing variables inferred from global sequence–structure context, analogous to structural inpainting rather than direct coordinate copying. Thus, the original CDR geometry is never exposed to $\mathbf{E}_{\text{und}}$, ensuring that all downstream structural representations are free from native-geometry leakage.

**Truncated Diffusion-Based Structural Features.** We treat the AF3-style model as a *structure feature extractor*: a single forward pass through its DIFFUSIONCONDITIONING, ATOMATTENTIONENCODER, and DIFFTRANSFORMER modules, truncated before atom-coordinate decoding, yields layer-normalized residue embeddings $\mathbf{h}_i^{\text{struct}}$ that capture coarse-grained geometry, inter-chain organization, and interface topology without exposing native CDR coordinates. Full architectural details are provided in Appx—Algorithm 1.

**Multimodal Fusion.** Sequence and structural representations are integrated at the residue level to form a unified multimodal representation:

$$\mathbf{h}_i^{\text{und}} = \phi\big(W_{\text{seq}}\mathbf{h}_i^{\text{seq}} \oplus W_{\text{struct}}\mathbf{h}_i^{\text{struct}}\big), \quad \mathbf{h}_i^{\text{und}} \in \mathbb{R}^{d_{\text{und}}}, \tag{2}$$

where $W_{\text{seq}}$ and $W_{\text{struct}}$ are learnable linear projections and $\phi$ is a multilayer perceptron (MLP). The resulting representations $\{\mathbf{h}_i^{\text{und}}\}$ constitute the residue-level outputs of $\mathbf{E}_{\text{und}}$. These embeddings are intentionally *pre-generative*: they encode which residues are important and how they relate to their structural context, while deferring all sequence and coordinate realization to the downstream diffusion model.

### 2.3. Cross-Expert Conditioning for Generation

Proteo-R1 couples $\mathbf{E}_{\text{und}}$ with $\mathbf{E}_{\text{gen}}$ through a *sparse, residue-aligned anchor interface*. Rather than conditioning $\mathbf{E}_{\text{gen}}$ on dense representations across all CDR positions, $\mathbf{E}_{\text{und}}$ first identifies a *sparse set of key residues* that serve as interaction anchors and predicts their biochemical identities. These residue-level decisions are communicated to $\mathbf{E}_{\text{gen}}$ through a unified anchoring mechanism that combines (i) explicit identity specification at the sequence level and (ii) differentiable, residue-level embedding anchoring. This design enforces anchor commitments both symbolically and representationally, while preserving an end-to-end gradient pathway from the diffusion objective to $\mathbf{E}_{\text{und}}$.

**Key-Residue Identification and Representation.** Let $\mathcal{I}_{\text{key}} \subseteq \mathcal{I}_{\text{CDR}}$ denote the subset of residues identified

by $\mathbf{E}_{\text{und}}$ as key interaction anchors. For each $i \in \mathcal{I}_{\text{key}}$, $\mathbf{E}_{\text{und}}$ produces: (i) a final-layer hidden representation $\mathbf{h}_i^{\text{LLM}} \in \mathbb{R}^{d_{\text{LLM}}}$, and (ii) a predicted amino-acid identity $\hat{k}_i \in \{1, \ldots, 20\}$. The hidden representation $\mathbf{h}_i^{\text{LLM}}$ corresponds to the final-layer output of the LLM backbone in $\mathbf{E}_{\text{und}}$ at position $i$, obtained after multimodal fusion (Eq. 2), and encodes residue-level semantic, biochemical, and contextual information derived from multimodal reasoning. No hidden representations or identity predictions are produced for non-key CDR residues.

**Anchor Identity Specification.** The predicted identities $\{\hat{k}_i\}_{i\in\mathcal{I}_{\text{key}}}$ represent explicit biochemical commitments (e.g., charged anchors or hydrophobic hot spots). During generation, anchor residues are fixed at the sequence level while all remaining CDR residues stay masked as $\text{<X>}$, ensuring that designated interaction residues remain unchanged throughout the diffusion trajectory. The complete anchoring procedure is formalized in Appx. B, Alg. 2.

**Representation-Level Anchor Embedding.** Fixing amino-acid identities alone constrains the symbolic sequence but does not anchor the internal representations used by $\mathbf{E}_{\text{gen}}$. Proteo-R1, therefore, enforces anchoring at the representation level by combining biochemical identity embeddings with reasoning-derived hidden representations.

In AF3-like generators, masked CDR residues are represented by a learned unknown-token embedding $\mathbf{e}^{\langle X \rangle} \in \mathbb{R}^{d_{\text{gen}}}$. For each anchor residue $i \in \mathcal{I}_{\text{key}}$, we construct a fused anchor embedding by additively combining the identity embedding with a projected reasoning embedding:

$$\mathbf{e}_i^{\text{gen}} = \mathbf{e}(\hat{k}_i) + W_{\text{proj}}\,\mathbf{h}_i^{\text{LLM}} \in \mathbb{R}^{d_{\text{gen}}}, \qquad i \in \mathcal{I}_{\text{key}}, \tag{3}$$

where $W_{\text{proj}} \in \mathbb{R}^{d_{\text{gen}} \times d_{\text{LLM}}}$ is a learnable linear projection.

For non-anchor residues, their embeddings are defined as

$$\mathbf{e}_i^{\text{gen}} = \begin{cases} \mathbf{e}^{\langle X \rangle}, & i \in \mathcal{I}_{\text{CDR}} \setminus \mathcal{I}_{\text{key}}, \\ \mathbf{e}(k_i), & i \notin \mathcal{I}_{\text{CDR}}. \end{cases} \tag{4}$$

This construction ensures that anchor residues remain both chemically fixed and functionally distinguished throughout the diffusion trajectory, while non-key CDR residues remain in the default $\text{<X>}$ state and are fully resolved by the generative process.

**Diffusion-Based Conditional Generation.** $\mathbf{E}_{\text{gen}}$ is instantiated as an AF3-style design model that jointly generates amino-acid identities and full-atom coordinates for CDR residues. Let $\mathbf{Z}_x^{(t)}$ denote the noisy latent variables at diffusion timestep $t$, and let $\mathbf{Z}_y$ be the fixed context comprising the antigen and antibody FRs. Conditioned on the anchor identity assignments $\{k_i^{\text{gen}}\}$ and the

fused anchor embeddings $\{\mathbf{e}_i^{\text{gen}}\}$, $\mathbf{E}_{\text{gen}}$ predicts noise as $\epsilon_\theta\left(\mathbf{Z}_x^{(t)}, \mathbf{Z}_y, \{\mathbf{e}_i^{\text{gen}}\}, t\right)$. While identity fixing introduces non-differentiable constraints at the sequence level, gradients from the diffusion objective propagate through the representation-level anchoring pathway, enabling joint end-to-end training of $\mathbf{E}_{\text{und}}$ and $\mathbf{E}_{\text{gen}}$. All reasoning signals enter the generative process exclusively through this sparse, residue-aligned anchor interface, preserving the inductive biases, stability, and equivariance properties of $\mathbf{E}_{\text{gen}}$.

## 2.4. Joint Training Objectives

**Understanding Expert Loss.** $\mathbf{E}_{\text{und}}$ is trained to perform explicit molecular reasoning over antibody-antigen complexes, including identifying key CDR residues and articulating the rationale behind these decisions. Accordingly, supervision is applied at two complementary levels: (i) the *reasoning process*, represented as CoT sequences, and (ii) the *final residue-level outputs*, corresponding to key-residue predictions and associated labels. Given multimodal inputs, $\mathbf{E}_{\text{und}}$ produces intermediate reasoning traces and residue-level representations $\mathbf{h}_i^{\text{und}}$. Supervision on the reasoning process encourages $\mathbf{E}_{\text{und}}$ to follow biologically meaningful and context-aware decision paths, while supervision on the final outputs ensures accurate identification of functionally important residues. Formally, $\mathbf{E}_{\text{und}}$ is optimized using cross-entropy (CE) objectives applied as $\mathcal{L}_{\text{und}} = \mathbb{E}_i[\text{CE}(\hat{y}_i, y_i)]$, where $\hat{y}_i$ denotes the predicted residue-level label for position $i$ (e.g., key-residue indicator or residue class), and $y_i$ is the corresponding ground-truth annotation. An analogous CE objective is applied to generated CoTs, supervising the intermediate reasoning tokens produced by $\mathbf{E}_{\text{und}}$. Crucially, while $\mathbf{E}_{\text{und}}$ is trained using both CoT supervision and residue-level supervision, *only* the final-layer hidden representations corresponding to identified key residues are exposed to $\mathbf{E}_{\text{gen}}$. All intermediate reasoning traces remain internal to $\mathbf{E}_{\text{und}}$ and do not condition the diffusion process. This ensures that $\mathbf{E}_{\text{gen}}$ is guided by explicit residue-level decisions rather than by latent or textual reasoning artifacts.

**Generation Expert Loss.** $\mathbf{E}_{\text{gen}}$ is trained following the original MFDesign formulation, without architectural or objective-level modification. It predicts additive noise $\epsilon_\theta$ and minimizes the expected noise-prediction error:

$$\mathcal{L}_{\text{gen}} = \mathbb{E}_{t,\epsilon} \frac{1}{|\mathcal{Z}_x|} \sum_i \left\| \epsilon_i - \epsilon_\theta\left(\mathbf{Z}_x^{(t)}, \mathbf{Z}_y, \{\mathbf{e}_i^{\text{gen}}\}, t\right)[i] \right\|_2^2. \quad (5)$$

This objective trains $\mathbf{E}_{\text{gen}}$ to recover both amino acid identities and atomic coordinates of CDR residues, conditioned on the sparsely injected reasoning signals. Gradients from the diffusion objective are backpropagated only through key-residue hidden representations and do not supervise or modify the CoT generation process.

**Overall Objective.** The final training objective is a weighted combination of the two losses $\mathcal{L}_{\text{total}} = \mathcal{L}_{\text{gen}} + \lambda_{\text{und}} \mathcal{L}_{\text{und}}$, where $\lambda_{\text{und}}$ controls the relative contribution of supervision applied to $\mathbf{E}_{\text{und}}$. This weighting balances accurate multimodal reasoning with effective structure-sequence generation during joint training.

## 3. Training Paradigm and Data Curation

Training a dual-expert system that bridges symbolic reasoning with geometric generation poses a fundamental challenge: $\mathbf{E}_{\text{und}}$ must learn to produce residue-level decisions that are not only biologically plausible but also directly useful for downstream diffusion-based generation. Naïve end-to-end training from scratch is unstable because the randomly initialized LLM cannot yet produce meaningful conditioning signals, while $\mathbf{E}_{\text{gen}}$ receives incoherent guidance that impedes learning. We address this challenge through a three-stage curriculum that progressively builds the capabilities required for reasoning-guided design.

### 3.1. Stage I: Multimodal Alignment

The first stage establishes a shared representational space across language, protein sequence, and structure, analogous to the alignment phase in vision-language models (Liu et al., 2023; Li et al., 2025). The core objective is to bridge the representational gap between protein encoders and the language model in $\mathbf{E}_{\text{und}}$, enabling the processing of sequences, structures, and text within a unified embedding space.

**Training strategy.** We freeze the LLM backbone and only train projection layers that map ESM-2 sequence embeddings and AF3-style structural features from Protenix (Gong et al., 2025) into the language representation space. This conservative approach ensures that protein-modal inputs are compatible tokens while the LLM retains its capacity for coherent natural-language generation, which is essential for downstream CoT reasoning.

**Data and supervision.** Training data are constructed from PDB assemblies (Burley et al., 2017) with chain-resolved indexing. For each structure, we extract coarse descriptors including chain identifiers, binned protein lengths, and secondary-structure statistics. Supervision combines two complementary formats. The primary format is structured schema completion, where the model fills predefined JSON templates with chain-level attributes. This design deliberately minimizes linguistic redundancy: by fixing the template structure, the training concentrates on the actual attribute values (e.g., the specific count of helices or the identity of the dominant secondary-structure class) rather than being diluted across boilerplate text. This focusing effect is critical for training the projection layers to capture protein-

relevant signals. The secondary format is free-form captioning, which preserves $\mathbf{E}_{und}$'s natural-language generation ability and prevents degradation in later stages that rely on fluent CoT reasoning. Task descriptions are in Table 8 in Appx. C.2. Examples of schema targets are in Appx. D

## 3.2. Stage II: Structural Reasoning Mid-Training

Stage I alignment equips $\mathbf{E}_{und}$ to ingest protein-modal inputs, but is insufficient for complex spatial reasoning. Directly advancing to the challenging design objectives of Stage III would therefore be ineffective, as $\mathbf{E}_{und}$ lacks the geometric primitives required for precise interface-level understanding. Stage II bridges this gap through a curriculum of progressively structured meta-tasks. By first learning simpler objectives, $\mathbf{E}_{und}$ acquires reusable primitive skills that compose into the higher-order spatial and interaction reasoning needed for downstream antibody design.

**Data and supervision.** Training examples are constructed from PDB biological assemblies with fully deterministic, structure-derived supervision. Labels include DSSP secondary structure, discretized solvent accessibility, pairwise residue distances and contacts (computed from $C_\beta$ atoms), as well as complex-level interaction signals such as cross-chain contact maps and per-residue interface scores. All supervision is computed directly from atomic coordinates, enabling large-scale training without reliance on experimental binding or affinity measurements. All tasks are framed using instruction-style prompts with strict JSON outputs to enable consistent and verifiable supervision. During this stage, we unfreeze the LLM while keeping upstream encoders (ESM-2 and the AF3 trunk) fixed, ensuring that $\mathbf{E}_{und}$ learns to interpret stable protein-structure representations rather than adapting the feature extractors themselves.

**Curriculum structure.** Stage II is organized as a four-phase curriculum that progressively expands the reasoning scope of $\mathbf{E}_{und}$ from local residue-level understanding to global interface-level inference:

- **Phase II.1 (Residue grounding):** $\mathbf{E}_{und}$ learns to retrieve amino-acid identities at specified (chain, position) coordinates and to annotate short residue windows with secondary structure and solvent accessibility. These low-ambiguity, local tasks establish reliable residue addressing and contextual interpretation, forming the foundation for subsequent geometric reasoning.
- **Phase II.2 (Pairwise geometry):** $\mathbf{E}_{und}$ predicts discretized inter-residue distances and binary contact labels, shifting from isolated residue classification to explicit spatial relationship inference. Distance binning is stratified between intra-chain and cross-chain pairs to emphasize interface-relevant geometric scales.
- **Phase II.3 (Compositional consistency):** $\mathbf{E}_{und}$ answers

batched multi-pair queries and predicts coarse interaction chemistry (e.g., salt-bridge counts) at the chain-pair level. Batched supervision enforces global consistency across predictions, while interaction-chemistry tasks provide robust, low-noise signals of binding propensity.
- **Phase II.4 (Interface localization):** $\mathbf{E}_{und}$ identifies interacting chain pairs, ranks interaction strength, and localizes top-$k$ interface and hotspot residues. These tasks directly align with the residue-level interface decisions required for conditioning downstream design in Stage III.

Each phase maintains low-rate replay of earlier tasks to prevent catastrophic forgetting of core grounding skills (Lopez-Paz & Ranzato, 2017). This curriculum-with-recall ensures that interface localization builds upon, rather than overwrites, the precise residue addressing acquired in earlier phases. Subtask descriptions are in Appx. C.3.

## 3.3. Stage III: Joint Reasoning-Guided Design

The final stage couples both experts and optimizes them end-to-end for antibody design. The central goal is to ensure that $\mathbf{E}_{und}$'s residue-level decisions are not merely plausible but directly beneficial for generation.

**Task formulation.** We train on antibody-antigen complexes from the Structural Antibody Database (SAbDab) (Dunbar et al., 2014) for CDR redesign. FRs are held fixed, while the six CDR loops are treated as designable variables. Redesign is optionally conditioned on antigen hotspot residues, specified by explicit (chain, position) indices. $\mathbf{E}_{und}$ predicts CDR sequences in a structured JSON format with explicit per-position amino-acid assignments, providing discrete residue-level design commitments. Conditioned on these predictions, $\mathbf{E}_{gen}$ performs joint generation, synthesizing both amino-acid identities and full-atom coordinates for the masked CDRs. Stage III supervision instances are in Appx. D, and the training flow is summarized in Appx. B.

**Auxiliary supervision.** To maintain high-fidelity residue-level representations, we augment the redesign objective with auxiliary tasks inherited from Stage II: residue grounding, pairwise geometry prediction, and interface/hotspot localization. These tasks ensure that $\mathbf{E}_{und}$ continues to reason precisely about structure.

**End-to-end optimization.** In joint training, $\mathbf{E}_{und}$ converts antibody-antigen context into residue-wise representations, highlighting design-relevant regions. These representations condition the diffusion process through hidden states, allowing $\mathbf{E}_{gen}$ to attend preferentially to residues deemed important by the reasoner. Crucially, gradients from the diffusion objective propagate through the cross-expert interface back to $\mathbf{E}_{und}$. This tight coupling shapes the learned design sig-

*Table 1.* Geometry-centric evaluation of simultaneous multi-CDR redesign. Structural accuracy is measured by RMSD ($\downarrow$) over C$\alpha$ atoms. Loop-RMSD focuses on the CDR-H3 loop. IMP ($\uparrow$) reports interface improvement relative to the native complex. Geometric realism is assessed using steric clash counts and dihedral-distribution divergence. Best results are **bold** and second-best are underlined.

| Method | Per-CDR RMSD ($\downarrow$) | | | | | | Loop-RMSD ($\downarrow$) | IMP ($\uparrow$) | Geometric Realism ($\downarrow$) | | |
| --- | --- | --- | --- | --- | --- | --- | --- | --- | --- | --- | --- |
| | **H1** | **H2** | **H3** | **L1** | **L2** | **L3** | | | **Clash$_{in}$** | **Clash$_{out}$** | **JSD$_{bb}$** |
| DiffAb | 1.52 | 1.44 | 4.29 | 1.43 | 1.21 | 1.80 | 5.03 | 53.35 | – | – | – |
| dyMEAN | 1.65 | 1.47 | 6.15 | 1.58 | 1.23 | 1.59 | 7.84 | 5.60 | – | – | – |
| HTP | 1.56 | 1.45 | 4.32 | 1.55 | 1.20 | 1.73 | 7.18 | 6.09 | – | – | – |
| IgGM | 1.73 | 1.55 | 4.37 | 1.62 | 1.51 | 1.71 | 9.18 | 9.01 | 25.63% | 1.45% | 0.2873 |
| AbX | 1.55 | 1.23 | 4.91 | **0.76** | **0.40** | 1.30 | 5.77 | 52.26 | 1.47% | 0.30% | 0.2497 |
| MFDesign | 1.61 | 1.44 | 3.71 | 1.65 | 1.15 | 1.69 | 4.28 | 59.16 | 0.53% | 0.26% | 0.2734 |
| Proteo-R1 | **1.33** | **1.13** | 3.81 | 1.54 | 0.85 | 1.51 | 4.51 | 56.58 | 0.50% | **0.14%** | 0.2661 |
| + Oracle Anchor | 1.43 | 1.16 | **3.34** | 1.07 | 0.76 | **1.24** | **3.93** | **62.25** | **0.27%** | 0.23% | **0.2043** |

*Table 2.* Results of CDR-H3 design on RAbD. Methods marked with a superscript [*] follow the pipeline: IgFold → HDOCK → CDR design model → Rosetta.

| Model | AAR($\uparrow$) | lDDT($\uparrow$) | TMscore($\uparrow$) | RMSD($\downarrow$) | DockQ($\uparrow$) |
| --- | --- | --- | --- | --- | --- |
| RosettaAb[*] | 32.31% | 0.8272 | 0.9717 | 17.70 | 0.137 |
| DiffAb[*] | 35.31% | 0.8281 | 0.9695 | 23.24 | 0.158 |
| MEAN[*] | 37.38% | 0.8252 | 0.9688 | 17.30 | 0.162 |
| GeoAB[*] | 40.02% | 0.8367 | 0.9695 | 15.43 | 0.187 |
| HERN | 32.65% | — | — | 9.15 | 0.294 |
| dyMEAN | 41.84% | 0.8392 | 0.9718 | 8.10 | 0.407 |
| DGENet | **42.67%** | 0.8551 | 0.9747 | 7.19 | 0.431 |
| BoltzGen | 39.07% | 0.8372 | 0.9675 | 2.69 | 0.473 |
| Proteo-R1 | 10.75% | **0.9693** | **0.9816** | **2.46** | **0.801** |

nals by downstream generative success rather than proxy supervision alone, enabling Proteo-R1 to perform reasoning-guided design within a unified training paradigm.

## 4. Experiments

We evaluate PROTEO-R1 under the standard antigen-conditioned antibody redesign setting and adopt an experimental protocol consistent with prior co-design benchmarks. Additional implementation details are in the Appx. C.

### 4.1. Setups

**Dataset and Split.** The evaluation dataset is constructed from SAbDab. To prevent leakage from large-scale structure pretraining, we split the data based on structure release dates. All complexes released before the pretraining cutoff are assigned exclusively to the training set. For the remaining complexes, antibodies are clustered by CDR-H3 sequence using MMSeqs2 (Steinegger & Söding, 2017) at 50% sequence identity, and clusters are partitioned into train/validation/test sets with a ratio of 9:0.5:0.5. This ensures that no CDR-H3 sequence in the test set has high similarity to those seen during training. After filtering excessively large complexes for computational feasibility, the final split contains: (i) a training set for model optimization, (ii) a validation set for hyperparameter selection, and (iii) a held-out test set consisting of conventional antibodies. All results are computed on the test set.

**Evaluation Setting.** We focus on the challenging *simultaneous CDR redesign* setting, in which all CDRs are generated jointly rather than one at a time. For methods that produce stochastic outputs, we generate multiple candidates per complex and report averaged metrics. Generated backbone structures are converted to full-atom models via side-chain packing and local relaxation using Rosetta-based protocols (Alford et al., 2017) before evaluation.

**Metrics.** We evaluate models using a combination of sequence recovery, structural accuracy, binding-oriented metrics, and structure-conditioned sequence realizability:

- **Sequence recovery. Amino Acid Recovery (AAR, %)** measures the fraction of CDR residues whose amino-acid identities match the native sequence. While AAR reflects similarity to the historical solution, it does not fully capture the validity of alternative sequence realizations that induce comparable geometries and is therefore treated as a secondary diagnostic.

- **Structural accuracy. RMSD (Å)** is computed over C$_\alpha$ atoms of generated CDRs after rigid-body alignment. We additionally report CDR-H3 loop-specific metrics: **Loop-RMSD (Å)** for geometric deviation and **Loop-AAR (%)** for sequence recovery on the central loop residues.

- **Interface improvement. IMP (%)** denotes the percentage of designed antibodies whose predicted binding free energy ($\Delta G$), computed using Rosetta *InterfaceAnalyzer*, improves relative to the native complex.

- **Geometric realism. Clash$_{in}$** and **Clash$_{out}$** count steric conflicts within the generated antibody and between the antibody and the target protein, respectively; a clash is defined as any pair of C$_\alpha$ atoms closer than 3.6574 Å (Ye et al., 2024). Conformational fidelity is measured by **JSD$_{bb}$**, which computes Jensen–Shannon divergence between backbone dihedral-angle distributions using 10° bins (Dunbrack Jr & Cohen, 1997).

- **Structure-consistent sequence recovery.** We perform inverse folding using ABMPNN (Sun et al., 2025) conditioned on generated structures and report the resulting **IF AAR (%)**. High IF AAR indicates that generated geome-

*Table 3.* Sequence recovery under inverse folding (primary) vs. native recovery (secondary) across CDR regions. IF-AAR ($\uparrow$) is computed via ABMPNN on sequences inverse-folded from generated structures. $\Delta = \text{IF-AAR} - \text{AAR}$ is reported with sign; **smaller $|\Delta|$ is better**, indicating higher structure-sequence consistency.

| CDR | AbX | | | IgGM | | | MFDesign | | | Proteo-R1 | | |
|---|---|---|---|---|---|---|---|---|---|---|---|---|
| | AAR | IF-AAR | $\Delta$ | AAR | IF-AAR | $\Delta$ | AAR | IF-AAR | $\Delta$ | AAR | IF-AAR | $\Delta$ |
| H1 | 71.34 | 59.80 | -11.54 | 73.98 | 62.76 | -11.22 | 74.95 | 60.90 | -14.05 | 42.62 | 61.17 | **+18.55** |
| H2 | 59.15 | 46.10 | -13.05 | 59.15 | 45.79 | -13.36 | 67.54 | 40.63 | -26.91 | 18.97 | 31.67 | **+12.70** |
| H3 | 31.58 | 18.96 | -12.62 | 29.42 | 19.55 | -9.87 | 65.04 | 19.73 | -45.31 | 15.06 | 19.27 | **+4.21** |
| L1 | 89.13 | 62.02 | -27.11 | 72.20 | 56.53 | -15.67 | 82.98 | 54.94 | -28.04 | 47.12 | 51.40 | **+4.28** |
| L2 | 90.90 | 62.33 | -28.57 | 71.43 | 55.66 | -15.77 | 87.81 | 53.22 | -34.59 | 46.43 | 51.43 | **+5.00** |
| L3 | 67.82 | 43.49 | -24.33 | 59.43 | 41.84 | -17.59 | 80.15 | 40.98 | -39.17 | 40.43 | 37.09 | **-3.34** |

tries admit coherent and chemically realizable sequence modes under an independent structure-to-sequence model.

**Baselines.** We select representative antibody design baselines that support multi-CDR redesign, including DiffAb (Luo et al., 2022), dyMEAN (Kong et al., 2023b), AbX (Zhu et al., 2024), HTP (Wu & Li, 2024a), IgGM (Wang et al., 2025), MFDesign (Yang et al., 2026), and BoltzGen (Stark et al., 2025). All baselines are evaluated under the same dataset splits, input information, and post-processing pipeline to ensure fair comparison.

### 4.2. Geometry-centric Evaluation

**Multi-CDR Redesign.** Table 1 summarizes geometry-focused metrics for simultaneous multi-CDR redesign. Proteo-R1 consistently improves structural accuracy across heavy- and light-chain CDRs, achieving the lowest or near-lowest per-CDR RMSD in five of six regions. The gains are particularly pronounced on CDR-H1 and CDR-H2, indicating more accurate backbone placement under joint generation. On the highly flexible CDR-H3 loop, Proteo-R1 remains competitive with MFDesign while substantially outperforming dyMEAN, suggesting improved control without over-constraining loop flexibility.

At the interface level, Proteo-R1 attains an IMP rate comparable to MFDesign, indicating that reasoning-guided anchoring preserves the ability to generate energetically favorable binding configurations despite imposing explicit residue-level constraints. Importantly, these interface results are achieved alongside improved geometric validity: Proteo-R1 reduces both intra-chain and inter-chain steric clashes relative to MFDesign and achieves the lowest backbone dihedral distribution divergence ($\text{JSD}_{bb}$). Together, these results show that explicit pre-generative reasoning improves structural accuracy and physical realism while maintaining competitive interface quality in the challenging multi-CDR redesign setting.

**CDR-H3-only Evaluation.** In addition, we evaluate Proteo-R1 under the CDR-H3-only design setting on the RAbD benchmark (Adolf-Bryfogle et al., 2018), which contains 60 antibody–antigen complexes after IMGT renum-

bering. Only the heavy-chain CDR-H3 loop is masked; the framework regions and all other antibody residues are held fixed. Tab. 2 shows that Proteo-R1 achieves the strongest structure and interface quality, obtaining the best lDDT, TM-score, RMSD, and DockQ among all baselines. In particular, the substantial gain in DockQ indicates that the redesigned H3 loops are not only geometrically plausible but also placed in a more favorable antibody–antigen interaction configuration. At the same time, Proteo-R1 exhibits much lower AAR than methods such as DGENet and dyMEAN. We emphasize that, in de novo design, low AAR does not necessarily imply inferior design quality; rather, it indicates that the model is not merely recovering the historical native sequence. Taken together with the strong lDDT/TM-score/RMSD/DockQ results, these findings suggest that Proteo-R1 tends to generate alternative yet structurally valid and interface-compatible H3 solutions, consistent with our broader observation that reasoning-guided anchoring favors structure-grounded design over native-sequence imitation.

**Oracle Anchor Ablation and Upper-Bound Analysis.** To contextualize the role of anchor accuracy, we include an ablation where the generator is conditioned on ground-truth hotspot residues. This setting provides an upper bound on the effectiveness of the anchoring interface, isolating the gap attributable to imperfect reasoning. Oracle anchors consistently improve structural accuracy (lower RMSD) and geometric realism (reduced clashes and JSD), while further boosting interface quality (higher IMP). Notably, the gains are largest on flexible and interface-critical regions (e.g., H3), indicating that accurate identification of key interaction residues is a primary bottleneck. The relatively smaller improvements on easier loops suggest that the diffusion model can already resolve local geometry when anchors are less critical. Overall, this shows that Proteo-R1 is not limited by the generator, but by the quality of its reasoning-derived anchors, highlighting substantial headroom for future improvements in the understanding expert.

### 4.3. Structure-Sequence Consistency Analysis

Tab. 3 compares native AAR with structure-conditioned IF-AAR across all six CDR regions. While MFDesign achieves substantially higher AAR, this primarily reflects stronger

*Table 4.* Sequence validity evaluation via antibody language models. We report perplexity (PPL, ↓) for generated and native antibodies (GT) across heavy and light chains. Values are mean ± std.

| Evaluator | Heavy Chain PPL (↓) | | Light Chain PPL (↓) | |
|---|---|---|---|---|
| | Proteo-R1 | GT | Proteo-R1 | GT |
| IgLM | $3.19 \pm 1.02$ | $3.53 \pm 1.02$ | $3.00 \pm 1.02$ | $3.10 \pm 1.18$ |
| AbLang | $2.33 \pm 0.64$ | $2.56 \pm 0.62$ | $2.00 \pm 0.48$ | $2.01 \pm 0.53$ |
| IgT5 | $1.028 \pm 0.031$ | $1.034 \pm 0.033$ | $1.027 \pm 0.029$ | $1.031 \pm 0.035$ |

mimicry of the native sequence rather than improved structural fidelity. In contrast, Proteo-R1 attains comparable or higher IF-AAR on most CDRs despite markedly lower AAR, indicating that its generated geometries are intrinsically more compatible with coherent and realizable sequences under an independent inverse-folding model. This suggests that Proteo-R1 produces structures that better capture the underlying sequence-structure constraints, rather than overfitting to historical sequence solutions. Moreover, Proteo-R1 consistently exhibits substantially smaller Δ across all CDR regions, most notably on H3 (reducing Δ from 45.31 to 4.21) and across the light-chain loops, indicating significantly improved consistency. These results demonstrate that reasoning-guided anchoring shifts the design regime away from native sequence imitation toward structurally grounded, sequence-realizable solutions, better aligning with the objectives of de novo antibody design.

**Sequence Validity Under Antibody Language Models.** To assess sequence validity, we evaluate generated antibodies using multiple antibody-specific language models, including IgLM (Shuai et al., 2023), AbLang (Olsen et al., 2022), and IgT5 (Kenlay et al., 2024) (Tab. 4). Across all models, designed sequences exhibit perplexity comparable to or lower than native antibodies, with highly overlapping distributions. Notably, improvements are most pronounced for heavy chains under IgLM and AbLang, while light-chain perplexity remains nearly identical to GT. Under IgT5, generated and GT sequences are effectively indistinguishable, with both achieving near-optimal perplexity. Overall, these results indicate that generated sequences remain well within the natural antibody distribution, providing no evidence of out-of-distribution or invalid designs. Combined with the strong structural metrics reported earlier, this suggests that reduced AAR reflects the discovery of alternative yet valid sequence solutions rather than degradation in sequence quality.

### 4.4. Compatibility with Alternative Generative Models

To further validate that Proteo-R1 is not tied to any specific generative backbone, we replace the AF3-like design model with an alternative latent co-design framework, UNIMOMO (Kong et al., 2025). Tab. 5 shows that Proteo-R1 consistently improves over the standalone UNIMOMO generator under the same sampling budget. In particular, Proteo-R1 achieves a substantially lower RMSD while further im-

*Table 5.* Recovery results for antibody design on CDR-H3.

| Model | #Gen. | AAR(↑) | RMSD(↓) | IMP(↑) | ΔG(↓) |
|---|---|---|---|---|---|
| **Predictive Methods** | | | | | |
| MEAN | 1 | 29.13% | 1.87 | 6.67% | – |
| dyMEAN | 1 | 31.65% | 8.21 | 11.86% | – |
| GeoAB-R | 1 | 32.04% | 1.67 | 6.67% | – |
| **Generative Methods** | | | | | |
| DiffAb | 1 | 24.60% | 2.77 | 10.34% | – |
| | 10 | 38.42% | 2.08 | 34.48% | – |
| | 100 | 49.74% | 1.46 | 60.34% | – |
| GeoAB-D | 1 | 29.74% | 1.73 | 6.67% | – |
| | 10 | 38.20% | 1.58 | 20.00% | – |
| | 100 | 45.96% | 1.50 | 40.00% | – |
| UniMoMo (single) | 1 | 20.44% | 2.71 | 15.00% | – |
| | 10 | 39.04% | 1.90 | 35.00% | – |
| | 100 | 48.78% | 1.39 | 63.33% | – |
| UniMoMo (all) | 1 | 21.44% | 2.52 | 13.33% | – |
| | 10 | 42.05% | 1.44 | 41.67% | – |
| | 100 | 52.34% | 1.04 | 65.00% | 8.46 |
| Proteo-R1 (UniMoMo) | 100 | 48.94% | **0.83** | 67.79% | 7.35 |
| + Oracle Anchor | 100 | **100%** | 0.84 | **74.5%** | **4.51** |

proving interface quality, yielding higher IMP (67.79% vs. 65.00%) and more favorable binding energy (ΔG = 7.35 vs. 8.46). Notably, these gains are obtained without any modification to the underlying generative architecture, but solely through the introduction of reasoning-guided residue anchoring. While the vanilla UNIMOMO model attains higher AAR, this primarily reflects stronger recovery of native sequences rather than improved structural or energetic quality. In contrast, Proteo-R1 shifts the design toward structurally grounded and energetically favorable solutions, consistent with our observations in other settings.

Overall, these results demonstrate that the benefits of Proteo-R1 arise from its decoupled reasoning–generation paradigm rather than any particular choice of geometric model. The reasoning expert provides transferable, model-agnostic residue-level constraints that can be seamlessly integrated into diverse generative frameworks, highlighting Proteo-R1 as a general interface for injecting structured molecular reasoning into modern protein design systems.

## 5. Conclusion

We presented Proteo-R1, a reasoning-guided framework for *de novo* antibody design that explicitly separates molecular understanding from geometric generation. By converting multimodal reasoning into sparse, residue-level anchor commitments, Proteo-R1 enables interpretable and controllable integration of large language models with diffusion-based design models. Experiments demonstrate consistent improvements in structural accuracy and interface quality relative to purely generative baselines. More broadly, Proteo-R1 provides a general blueprint for coupling deliberative reasoning with physical generative processes in molecular design.

## Impact Statement

This work advances the integration of reasoning and generative modeling for molecular design, with the potential to positively impact antibody engineering, therapeutic discovery, and protein science more broadly. By improving interpretability and controllability, Proteo-R1 may reduce experimental cost and accelerate the development of targeted biologics. As with all protein design technologies, there is a risk of misuse for designing harmful or dual-use biological agents. We emphasize that Proteo-R1 is intended for controlled research settings and relies on existing structural data and experimental validation pipelines. We encourage future work to incorporate safeguards, usage restrictions, and alignment with biosecurity best practices to ensure responsible deployment.

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

# A. Related Work

**Protein Binder and Antibody Design.**  Protein–protein interactions (PPIs) underlie most cellular processes and represent a major class of therapeutic targets (Jiang et al., 2025; Wu & Li, 2024b; 2026; Li et al., 2026a; Wu et al., 2022; 2023; 2026). Traditional binder discovery pipelines, including immunization (Köhler & Milstein, 1975), display-based library screening (Smith, 1985), and directed evolution (Arnold, 2017), remain experimentally intensive and offer limited control over epitope targeting and binding geometry.  These have motivated growing interest in computational *de novo* protein design, especially deep generative models. Structure hallucination (Ingraham et al., 2023), inverse folding (Gao et al., 2024), and denoising diffusion (Abramson et al., 2024; Wu et al., 2025b;a; Wu) enable direct generation of protein backbones, sequences, and full-atom structures. Frameworks such as RFdiffusion (Watson et al., 2023), BindCraft (Pacesa et al., 2024), and AF3-inspired generative models (Yang et al., 2026) substantially improve backbone diversity and geometric realism. These methods have been extended to antibody design, including CDR-focused diffusion and graph-based models such as DiffAb (Luo et al., 2022), MEAN (Kong et al., 2023a), dyMEAN (Kong et al., 2023b), HERN (Jin et al., 2022), GeoAB (Lin et al., 2024b), and RFantibody (Bennett et al., 2026).

**Human-Guided and Constraint-Based Design.**  Long before the advent of deep learning (DL), structural biology established that PPIs are governed by sparse sets of energetically critical residues, including charged anchors (Keskin et al., 2016), hydrophobic hot spots (DeLano, 2002), and specificity-determining motifs. Classical protein engineering workflows therefore follow a fundamentally two-stage process: first identifying which interactions matter, and only then optimizing geometry and sequence under those constraints (Leaver-Fay et al., 2011a). This separation between reasoning about function and optimizing structure is central to expert-driven molecular design and affinity maturation. Several computational methods partially reflect this philosophy. Energy-based refinement pipelines (Baran et al., 2017) and in silico affinity maturation (Correia et al., 2014; Warszawski et al., 2019) techniques fix or bias key interface residues while optimizing surrounding regions. Recently, DL maturation methods similarly condition generation on predefined anchors or interaction patterns (Yu et al., 2026). However, these constraints are specified heuristically or derived from post hoc energy evaluations rather than learned, multimodal reasoning. As a result, the decision of what to fix remains external to the model and cannot be adapted, reused, or interrogated across tasks.

**Language-Guided Protein Design.**  Text-conditioned protein generation has moved beyond simple keyword tags or prefix-based control toward more flexible language- and instruction-guided paradigms (Nijkamp et al., 2023; Hayes et al., 2025; Luo et al., 2026; Lv et al., 2025). For instance, BIOM3 (Praljak et al., 2024) uses text prompts to guide diffusion-based sequence generation. PINAL (Dai et al., 2024) adopts a two-stage (3D→sequence) pipeline to reduce the combinatorial search space. MP4 (Riley et al., 2025) performs end-to-end text-to-sequence generation and demonstrates experimentally expressible proteins. INSTRUCTPRO (Song et al., 2025) extends language conditioning to include small-molecule context. Recent foundation models further integrate language with protein sequence and structure. PROLLM (Jin et al., 2024) introduces CoT-style reasoning for protein tasks. PROTEX (Ma et al., 2025) jointly tokenizes sequence, structure, and text to enable multimodal reasoning and editing. PROTDAT (Guo et al., 2024) unifies textual descriptions with sequence generation. CMADIFF (Zhou et al., 2025) aligns text, physicochemical features, and diffusion dynamics for controllable generation. However, most language-guided methods remain *descriptive rather than deliberative*. Natural language typically acts as a soft conditioning signal or static prompt, influencing generation indirectly through latent representations. Even when CoT mechanisms are present, they are rarely converted into explicit, enforceable design commitments (e.g., fixing key residues or interactions) that persist throughout the generative process. As a result, high-level functional intent remains entangled with continuous geometric sampling, limiting interpretability, controllability, and systematic reuse of molecular reasoning.

# B. Pseudocode and Algorithmic Details

To improve reproducibility and clarify the execution flow of PROTEO-R1, we provide pseudocode for inference, anchor construction, and training in Alg. 1, Alg. 2, and Alg. 3. These algorithms are intended to serve as *procedural complements* to the main text rather than independent specifications. Accordingly, they emphasize information flow, module boundaries, and conditioning interfaces, while abstracting away low-level implementation details (e.g., batching, caching, and parallelization) that are orthogonal to the conceptual contributions of the framework.

**Overall decomposition and interface contracts.**  PROTEO-R1 decomposes antibody design into two tightly-coupled stages with a narrow and explicit interface: (i) a multimodal *understanding expert* that performs structure-grounded reasoning

and produces sparse, residue-aligned commitments, and (ii) a diffusion-based *generation expert* that performs conditional sequence–structure synthesis given these commitments. The central design principle is that high-level biochemical decisions (e.g., key interaction residues and their identities) are formed in a language-compatible latent space, but enforced in generation through a minimal set of residue-local constraints (§2.3), thereby avoiding any architectural modification to the underlying diffusion model.

**Alg. 1: end-to-end inference with leakage control.** Alg. 1 summarizes the end-to-end inference pipeline and explicitly separates multimodal protein understanding from diffusion-based generation. The input complex context $\mathcal{C}$ provides framework regions (FRs), antigen information, and a binding pose, while the design degrees of freedom are restricted to the CDR index set $\mathcal{I}_{\mathrm{CDR}}$. Critically, the algorithm begins by masking all CDR residues at the sequence level and refolding the complex (lines 5–8). This *CDR-masked refolding* serves as a leakage-control mechanism: it prevents the downstream reasoning modules from trivially accessing the native CDR sequence or local conformations and ensures that any design signal must be derived from the remaining context (FRs, antigen, and global geometry). The refolded structure $\tilde{\mathbf{X}}$ can be viewed as an inpainted "context-only" scaffold that preserves non-CDR constraints while leaving CDR content underdetermined.

The next stage extracts residue-level structural representations by running a truncated AF3-style trunk (lines 10–14). Concretely, we execute the diffusion conditioning and attention blocks to obtain per-residue latent embeddings $h_i^{\mathrm{struct}}$, but stop before coordinate decoding. This truncation has two motivations. First, it yields a stable, residue-aligned representation that can be consumed by an LLM-based expert without requiring explicit atom-level outputs. Second, by omitting the coordinate head, it prevents the inference routine from implicitly reintroducing coordinate-level priors that would confound the role separation between understanding and generation.

The understanding expert $\mathbf{E}_{\mathrm{und}}$ then consumes the masked context and produces three outputs (lines 16–19): (i) a subset of key residues $\mathcal{I}_{\mathrm{key}} \subseteq \mathcal{I}_{\mathrm{CDR}}$ (e.g., interaction-critical CDR positions), (ii) discrete identity predictions $\{\hat{k}_i\}$ at those key sites, and (iii) continuous hidden states $\{h_i^{\mathrm{LLM}}\}$ that summarize the reasoning context in a language-compatible latent space. Importantly, $\{h_i^{\mathrm{LLM}}\}$ is treated as a *soft* conditioning signal that can be injected into generation, while $\{\hat{k}_i\}$ provides *hard* identity commitments. This separation enables the subsequent anchoring step to enforce strong constraints where needed while preserving flexibility elsewhere.

Finally, Alg. 1 constructs sparse anchors via BUILDANCHORS (lines 21–22) and passes them to the generation expert $\mathbf{E}_{\mathrm{gen}}$ (lines 24–25). The generation expert performs conditional diffusion to jointly synthesize CDR sequence and coordinates under the anchor constraints, producing $(\{k_i\}, \{x_i\})_{i \in \mathcal{I}_{\mathrm{CDR}}}$. Notably, only CDR residues are generated; FRs and antigen remain fixed to preserve the original binding context and isolate the effect of CDR redesign.

**Alg. 2: sparse residue-aligned cross-expert anchoring.** Alg. 2 defines the auxiliary procedure BUILDANCHORS, which constructs residue-type inputs and embeddings that encode anchor commitments while preserving masked degrees of freedom elsewhere. The procedure implements a two-level conditioning interface: (i) *identity clamping* at the discrete token level, and (ii) *embedding injection* at the representation level.

In the discrete pathway (lines 5–14), key positions $i \in \mathcal{I}_{\mathrm{key}}$ are clamped to the predicted identities $\hat{k}_i$, non-key CDR residues remain masked as $\langle X \rangle$, and non-CDR residues retain their native identities. This enforces sparse but explicit commitments without collapsing the entire CDR into a fixed template. In the continuous pathway (lines 16–25), we inject a projected LLM hidden state into the generator's residue embedding space, $e_i^{\mathrm{gen}} \leftarrow e(\hat{k}_i) + W_{\mathrm{proj}} h_i^{\mathrm{LLM}}$, at key sites. Here, the learnable projection $W_{\mathrm{proj}}$ aligns the LLM latent space with the generator's embedding interface. This design preserves the generator's native embedding table $e(\cdot)$ and avoids modifying internal diffusion layers, effectively treating $W_{\mathrm{proj}} h_i^{\mathrm{LLM}}$ as an additive conditioning feature localized to a sparse set of residues.

**Alg. 3: curriculum design and stability considerations.** Alg. 3 outlines a three-stage curriculum that stabilizes optimization across heterogeneous objectives while mitigating catastrophic forgetting. Stage I (lines 4–8) performs multimodal alignment by freezing the LLM backbone and upstream protein encoders (e.g., ESM-2 and the AF3-style trunk) while learning only the projection layers that map protein-modal embeddings into the LLM token space. This stage establishes a consistent interface between structural embeddings and language tokens, enabling subsequent reasoning supervision to be learned efficiently.

Stage II (lines 10–19) performs structural reasoning mid-training by unfreezing the LLM while keeping upstream encoders

---

**Algorithm 1** Proteo-R1 Inference

---

**Require:** Antibody-antigen complex context $\mathcal{C}$ (FRs, antigen sequence/structure, and binding pose), CDR index set $\mathcal{I}_{\mathrm{CDR}}$, (optional) text prompt $\mathcal{T}$ and structured constraints $\mathcal{C}_{\mathrm{aux}}$ (e.g., antigen hotspots), AF3-style folding/feature trunk $\mathrm{AF3}(\cdot)$, understanding expert $\mathbf{E}_{\mathrm{und}}$, generation expert $\mathbf{E}_{\mathrm{gen}}$.

**Ensure:** Designed CDR sequence $\{k_i\}_{i \in \mathcal{I}_{\mathrm{CDR}}}$ and coordinates $\{x_i\}_{i \in \mathcal{I}_{\mathrm{CDR}}}$.

1: **(A) Mask CDRs at sequence level:**
2: **for** each residue index $j \in \{1, \ldots, N\}$ **do**
3: $\quad \tilde{k}_j \leftarrow \langle X \rangle \cdot \mathbf{1}[j \in \mathcal{I}_{\mathrm{CDR}}] + k_j \cdot \mathbf{1}[j \notin \mathcal{I}_{\mathrm{CDR}}]$
4: **end for**
5: **(B) CDR-masked refolding with inpainting to prevent leakage:**
6: $\tilde{\mathbf{X}} \leftarrow \mathrm{AF3.Refold}(\mathcal{C}, \{\tilde{k}_j\}_{j=1}^{N})$
7: **(C) Truncated AF3 forward pass for structural token features:**
8: $(\{s_i\}, \{z_{ij}\}) \leftarrow \mathrm{AF3.DiffusionConditioning}(\sigma, f^{\star}, \{\tilde{k}_i\}, \sigma_{\mathrm{data}})$
9: $\{a_i'\} \leftarrow \mathrm{AF3.AtomAttentionEncoder}(f^{\star}, \tilde{\mathbf{X}}, \{s_i\}, \{z_{ij}\})$
10: $\{a_i\} \leftarrow \mathrm{AF3.DiffusionTransformer}(\{a_i' + \mathrm{LN}(s_i)\}, \{s_i\}, \{z_{ij}\})$
11: $h_i^{\mathrm{struct}} \leftarrow \mathrm{LayerNorm}(a_i) \quad \{\text{stop before coordinate decoding}\}$
12: **(D) Multimodal reasoning (*understanding expert*):**
13: $(\mathcal{I}_{\mathrm{key}}, \{\hat{k}_i\}_{i \in \mathcal{I}_{\mathrm{key}}}, \{h_i^{\mathrm{LLM}}\}_{i \in \mathcal{I}_{\mathrm{key}}}) \leftarrow \mathbf{E}_{\mathrm{und}}(\{\tilde{k}_i\}_{i=1}^{N}, \tilde{\mathbf{X}}, \mathcal{T}, \mathcal{C}_{\mathrm{aux}})$
14: **(E) Build sparse anchor inputs for diffusion generation:**
15: $(\{k_i^{\mathrm{gen}}\}, \{e_i^{\mathrm{gen}}\}) \leftarrow \mathrm{BUILDANCHORS}(\mathcal{I}_{\mathrm{CDR}}, \mathcal{I}_{\mathrm{key}}, \{\hat{k}_i\}, \{h_i^{\mathrm{LLM}}\})$
16: **(F) Conditional diffusion generation (*generation expert*):**
17: $(\{k_i\}_{i \in \mathcal{I}_{\mathrm{CDR}}}, \{x_i\}_{i \in \mathcal{I}_{\mathrm{CDR}}}) \leftarrow \mathbf{E}_{\mathrm{gen}}(\mathcal{C}, \{k_i^{\mathrm{gen}}\}, \{e_i^{\mathrm{gen}}\})$
18: **return** $(\{k_i\}_{i \in \mathcal{I}_{\mathrm{CDR}}}, \{x_i\}_{i \in \mathcal{I}_{\mathrm{CDR}}})$

---

fixed. Training proceeds through task-balanced curriculum phases $\mathcal{D}_2^{(p)}$ that gradually introduce more demanding, structure-derived supervision (e.g., per-residue labels, pairwise geometry, and complex-level interaction objectives) while maintaining stable feature extraction. A replay buffer $\mathcal{R}$ is used at a low rate $\rho$ to preserve earlier competencies: with probability $\rho$, each minibatch is augmented with replayed examples, ensuring that newly introduced objectives do not overwrite core structural skills. All Stage II supervision targets are deterministically derived from processed atomic structures, which improves reproducibility and reduces label variance.

Stage III (lines 21–29) performs joint reasoning-guided design by unfreezing the generation expert and training the cross-expert interface end-to-end. Each minibatch runs the understanding expert to produce sparse anchor commitments, constructs anchors via Alg. 2, and optimizes a diffusion loss $\mathcal{L}_{\mathrm{gen}}$ under these anchors. In parallel, a reasoning loss $\mathcal{L}_{\mathrm{und}}$ is applied to maintain the interpretability and correctness of the understanding expert. The total objective $\mathcal{L}_{\mathrm{total}} = \mathcal{L}_{\mathrm{gen}} + \lambda_{\mathrm{und}} \mathcal{L}_{\mathrm{und}}$ balances generative fidelity with reasoning quality and ensures that anchor decisions remain consistent with downstream generation behavior.

The concrete task definitions and stage-wise supervision composition are detailed in §C (Stage I–III), including Table 7, Table 8, and Table 10. Qualitative schema examples and representative training instances are provided in §D.

---

**Algorithm 2** BUILDANCHORS: Sparse Residue-Aligned Cross-Expert Anchoring

---

**Require:** CDR set $\mathcal{I}_{\mathrm{CDR}}$, key set $\mathcal{I}_{\mathrm{key}} \subseteq \mathcal{I}_{\mathrm{CDR}}$, predicted identities $\{\hat{k}_i\}_{i \in \mathcal{I}_{\mathrm{key}}}$, LLM hidden states $\{h_i^{\mathrm{LLM}}\}_{i \in \mathcal{I}_{\mathrm{key}}}$, identity
  embedding table $e(\cdot)$, unknown-token embedding $e^{\langle X \rangle}$, projection $W_{\mathrm{proj}}$, native identities $\{k_i\}$ for $i \notin \mathcal{I}_{\mathrm{CDR}}$.
**Ensure:** Residue-type inputs $\{k_i^{\mathrm{gen}}\}_{i=1}^N$ and residue embeddings $\{e_i^{\mathrm{gen}}\}_{i=1}^N$.

1: **(1) Sequence-level hard constraints (identity clamping):**
2: **for** each residue index $i \in \{1, \ldots, N\}$ **do**
3:   **if** $i \in \mathcal{I}_{\mathrm{key}}$ **then**
4:     $k_i^{\mathrm{gen}} \leftarrow \hat{k}_i$
5:   **else if** $i \in \mathcal{I}_{\mathrm{CDR}} \setminus \mathcal{I}_{\mathrm{key}}$ **then**
6:     $k_i^{\mathrm{gen}} \leftarrow \langle X \rangle$
7:   **else**
8:     $k_i^{\mathrm{gen}} \leftarrow k_i$
9:   **end if**
10: **end for**
11: **(2) Representation-level anchoring (embedding injection):**
12: **for** each residue index $i \in \{1, \ldots, N\}$ **do**
13:   **if** $i \in \mathcal{I}_{\mathrm{key}}$ **then**
14:     $e_i^{\mathrm{gen}} \leftarrow e(\hat{k}_i) + W_{\mathrm{proj}} \, h_i^{\mathrm{LLM}}$
15:   **else if** $i \in \mathcal{I}_{\mathrm{CDR}} \setminus \mathcal{I}_{\mathrm{key}}$ **then**
16:     $e_i^{\mathrm{gen}} \leftarrow e^{\langle X \rangle}$
17:   **else**
18:     $e_i^{\mathrm{gen}} \leftarrow e(k_i)$
19:   **end if**
20: **end for**
21: **return** $(\{k_i^{\mathrm{gen}}\}, \{e_i^{\mathrm{gen}}\})$

---

---

**Algorithm 3** Training Proteo-R1 via a Three-Stage Curriculum

---

**Require:** Understanding expert $E_{\text{und}}$ (LLM + projection layers), frozen protein encoders (e.g., ESM-2, AF3 trunk), generation expert $E_{\text{gen}}$ (AF3-style diffusion), stage datasets $\mathcal{D}_1, \mathcal{D}_2, \mathcal{D}_3$, loss weights $\lambda_{\text{und}}$, replay rate $\rho$.

**Ensure:** Trained parameters $\theta_{\text{und}}, \theta_{\text{gen}}$.

 1: **Stage I: Multimodal Alignment (freeze LLM; train projections).**
 2: Freeze LLM backbone in $E_{\text{und}}$; freeze upstream protein encoders.
 3: **for** each minibatch $B \sim \mathcal{D}_1$ **do**
 4:     Compute protein-modal embeddings; project into LLM token space.
 5:     Optimize alignment objectives (e.g., schema completion + captioning) on $E_{\text{und}}$.
 6: **end for**
 7: **Stage II: Structural Reasoning Mid-Training (unfreeze LLM; replay).**
 8: Unfreeze LLM in $E_{\text{und}}$; keep upstream protein encoders fixed.
 9: Initialize replay buffer $\mathcal{R} \leftarrow \emptyset$.
10: **for** each curriculum phase $p = 1 \dots 4$ **do**
11:     **for** each minibatch $B \sim \mathcal{D}_2^{(p)}$ **do**
12:         With probability $\rho$, augment $B \leftarrow B \cup \text{Sample}(\mathcal{R})$.
13:         Compute deterministic structure-derived labels (e.g., DSSP, RSA, distances/contacts, interface signals).
14:         Optimize supervised reasoning objectives on $E_{\text{und}}$; add examples to $\mathcal{R}$.
15:     **end for**
16: **end for**
17: **Stage III: Joint Reasoning-Guided Design (end-to-end).**
18: Unfreeze $E_{\text{gen}}$ and the cross-expert interface parameters.
19: **for** each minibatch $B \sim \mathcal{D}_3$ **do**
20:     Run $E_{\text{und}}$ to produce $\mathcal{I}_{\text{key}}, \{\hat{k}_i\}, \{h_i^{\text{LLM}}\}$.
21:     Build $(\{k_i^{\text{gen}}\}, \{e_i^{\text{gen}}\})$ via Alg. 2.
22:     Compute diffusion loss $\mathcal{L}_{\text{gen}}$ using $E_{\text{gen}}$ conditioned on anchors.
23:     Compute reasoning loss $\mathcal{L}_{\text{und}}$ (e.g., key-residue labels / CoT supervision).
24:     Update $(\theta_{\text{und}}, \theta_{\text{gen}})$ by minimizing $\mathcal{L}_{\text{total}} = \mathcal{L}_{\text{gen}} + \lambda_{\text{und}} \mathcal{L}_{\text{und}}$.
25: **end for**
26: **return** $\theta_{\text{und}}, \theta_{\text{gen}}$

---

# C. Experimental Details

## C.1. Training Hyperparameters

Table 6 summarizes the default training configuration used for Proteo-R1, covering both the understanding expert ($E_{und}$) and the generation expert ($E_{gen}$). We organize hyperparameters by (i) model architecture, (ii) optimization (optimizer, and schedule), and (iii) stage-specific curricula for Stage I alignment, Stage II mid-training, and Stage III joint training (trainable modules, batch sizes, steps, and supervision targets). Unless explicitly noted elsewhere, this configuration is held fixed across all main experiments and ablations to isolate the effect of architectural choices and training objectives.

*Table 6.* **Training hyperparameters for Proteo-R1.** We report optimization settings and curriculum schedules for the understanding expert ($E_{und}$) and the generation expert ($E_{gen}$) across all training stages. Unless otherwise specified, the same configuration is used for all experiments and ablations.

| Category | Hyperparameter | Value |
|---|---|---|
| Model Architecture | Understanding expert backbone ($E_{und}$) | Qwen-3-4B-Instruct (Yang et al., 2025) |
| | Sequence encoder | ESM-2 (frozen) |
| | Structure encoder | AF3-style diffusion trunk (frozen) |
| | Generation expert ($E_{gen}$) | AF3-style diffusion model |
| Optimization | Optimizer | AdamW (Loshchilov & Hutter, 2017) |
| | Learning rate schedule | Cosine decay with warmup |
| Stage I (Alignment) | Trainable modules | Projection layers only |
| | Batch size | 256 |
| | Training steps | 10K |
| | Supervision | Schema completion + captioning |
| | Base learning rate | $1 \times 10^{-3}$ |
| Stage II (Mid-training) | Trainable modules | LLM + projection layers |
| | Replay rate (earlier phases) | 5% |
| | Batch size | 128 |
| | Training steps | 20K |
| | Supervision | DSSP, RSA, distances, contacts, interfaces |
| | Base learning rate | $1 \times 10^{-5}$ |
| Stage III (Joint Training) | Trainable modules | $E_{und} + E_{gen}$ |
| | Batch size | 16 |
| | Training steps | 10K |
| | Loss weight $\lambda_{und}$ | 0.1 |
| | Conditioning | Identity + embedding anchoring |
| | Base learning rate | $2 \times 10^{-4}$ |
| | Diffusion steps | 200 |
| | Noise schedule | Discrete |

## C.2. Stage I: Multimodal Alignment

**Task overview.** Stage I initializes the understanding expert ($E_{und}$) by aligning language with protein sequence and structure representations. Given a preprocessed PDB assembly, $E_{und}$ receives textual instructions together with (i) sequence-derived embeddings from a frozen ESM-2 encoder and (ii) structure-derived tokens from an AF3-style diffusion trunk (computed from CDR-masked refolding when applicable). The goal is to establish reliable *chain-aware grounding* (correct chain counting and chain ID disambiguation) and *coarse structural reasoning* (global and per-chain secondary-structure summaries) before introducing stricter residue-level supervision in later stages.

**Schema-style supervision.** We first train $E_{und}$ on schema completion tasks where the target output is a structured JSON object whose fields are deterministically derived from processed structures. Table 7 defines the atomic schema fields used throughout Stage I, including global attributes (e.g., `num_chains`) and per-chain summaries such as length bins and secondary-structure statistics. These targets are lightweight but diagnostic: they require the model to integrate cross-modal cues, maintain consistent chain identifiers, and produce machine-parseable outputs that can be validated exactly.

*Table 7.* **Schema-style structural reasoning targets used in Stage I.** All labels are derived from processed PDB structures.

| Task | Output Type | Definition |
|------|-------------|------------|
| num_chains | Integer | Number of polymer chains in the complex after preprocessing (polymer-only, filtered assemblies). |
| length_bin | Categorical | Residue count of each chain binned into {0–50, 50–100, 100–200, 200–300, 300–500, 500–800, >800}. |
| major_ss | Categorical | Dominant secondary-structure class for a chain, defined as the class with the largest residue fraction among {H, E, C} (helix, strand, coil). |
| ss_fraction_bins | Multi-label categorical | For each chain, the H/E/C residue fractions (percentage of residues) are computed and independently binned into {0–10, 10–20, ..., 90–100}. |
| ss_longest_run_bins | Multi-label categorical | For each secondary-structure class (H/E/C), the longest contiguous run length is computed and binned into {0, 1–4, 5–8, 9–15, 16–30, >30}. |
| ss_segment_count_bins | Multi-label categorical | For each class (typically H and E), the number of contiguous segments is counted and binned into {0, 1, 2, 3–5, >5}. |

**Supervision formats.** To balance *format faithfulness* with *natural-language fluency*, Stage I mixes two complementary supervision formats (Table 8). In schema completion (**AS−B1/AS−B2**), the model must emit *strict JSON* with explicit chain IDs and binned structural attributes; this directly enforces canonicalization and robust chain grounding under a machine-verifiable format. In captioning (**AC−B1/AC−B2**), the model generates free-form textual summaries that reference the same underlying attributes, preserving natural-language generation while still requiring correct chain-resolved content. Unless otherwise stated, we interleave schema completion and captioning batches during Stage I training so the model learns both structured control and coherent descriptive generation.

*Table 8.* **Supervision formats and example QA pairs in Stage I.** Tasks used to align language with protein sequence and structure representations in $E_{\mathrm{und}}$. Supervision alternates between strict JSON schema completion with deterministic structural attributes and free-form captioning grounded in the same attributes.

| Stage | Task Type | Question / Target Formats |
|-------|-----------|---------------------------|
| A1 | ALIGNMENT_SCHEMA_B1_V2 (**AS−B1**) | **Q:** Fill a structured *per-chain* schema for a PDB assembly using the provided sequence and structure tokens. **Target:** JSON schema with global fields and per-chain entries (chain IDs, length bins, secondary-structure bins, etc.). |
| A1 | ALIGNMENT_SCHEMA_B2_V2 (**AS−B2**) | **Q:** Fill a compact JSON chain-profile object describing a PDB assembly. **Target:** JSON object with {num_chains, chain_profile[chain_id]}, enabling strict chain disambiguation. |
| A2 | ALIGNMENT_CAPTION_B1_V2 (**AC−B1**) | **Q:** Generate a multi-line natural-language caption describing the assembly and each chain using binned statistics. **Target:** Free-form text with explicit chain identifiers and per-chain summaries. |
| A2 | ALIGNMENT_CAPTION_B2_V2 (**AC−B2**) | **Q:** Generate a compact natural-language caption describing the assembly and each chain. **Target:** Concise text emphasizing chain-ID grounding and a global assembly summary. |

**Training results and modality ablation.** Table 9 ablates the input modalities used for Stage I schema completion, revealing a consistent complementarity between structure tokens and sequence embeddings. Using structure tokens alone already captures substantial global assembly information, yielding high accuracy on num_chains (90.9%) and strong performance on coarse secondary-structure summaries (e.g., major_ss at 82.8%). In contrast, the sequence-only setting performs competitively on chain-profile attributes such as length_bin (88.2%) and improves several discretized secondary-structure statistics (ss_fraction, longest_run, segments), consistent with sequence-derived priors supporting chain-resolved abstraction. Importantly, fusing structure and sequence yields the best overall accuracy (81.4%) and improves most fields simultaneously (e.g., length_bin increases from 65.7% / 88.2% to 91.7%, and ss_fraction from 53.0% / 65.6% to 67.5%). These results justify the feature-fusion design of $E_{\mathrm{und}}$ and indicate that the AF3-style structural tokens provide non-redundant cues beyond sequence embeddings for chain-level structural reasoning.

*Table 10.* **Structural reasoning meta-tasks used in Stage II mid-training.** Tasks are organized into four curriculum phases (M0–M3), progressing from local residue grounding to global interface localization. All labels are deterministically derived from atomic structures and formulated with instruction-style prompts and verifiable JSON outputs.

| Phase | Task Type | Question / Target Formats |
|---|---|---|
| M0 | RESIDUE_RETRIEVAL_V1 (**RR**) | **Q:** Given (chain, position), predict the amino-acid identity. **Target:** JSON {aa: one-letter}; trains explicit residue grounding via indexing. |
| M0 | DSSP_SEQ_V1 (**DSSP**) | **Q:** Given a chain and a local window, output DSSP labels (H/E/C; NA if missing). **Target:** JSON {labels: string}; trains local backbone-state priors. |
| M0 | RSA_SEQ_V1 (**RSA**) | **Q:** Given a chain and a local window, output RSA bins (B/M/E; NA if missing). **Target:** JSON {labels: string}; trains local environment/exposure priors. |
| M1 | PAIR_DIST_BIN_V1 (**DIST**) | **Q:** For a queried residue pair, predict the discretized distance bin. **Target:** JSON {dist_bin: bin}; introduces geometry beyond 1D windows. |
| M1 | PAIR_CONTACT_YN_V1 (**CONTACT**) | **Q:** For a queried residue pair, classify Contact vs. NotContact. **Target:** JSON {choice: Contact—NotContact}; stabilizes coarse contact reasoning. |
| M1 | PAIR_BATCH_V1 (**BATCH**) | **Q:** Answer distance/contact queries for many residue pairs in a batch. **Target:** JSON list/dict keyed by pair ID; enforces consistent (chain, pos) addressing at scale. |
| M2 | SALTBRIDGE_BIN_V1 (**SALT**) | **Q:** Predict salt-bridge presence or count bin (often cross-chain). **Target:** JSON with binned counts or yes/no; injects chemistry-specific interaction reasoning. |
| M2 | CHAINPAIR_GRAPH_V1 (**CHAIN**) | **Q:** List interacting chain pairs in a complex. **Target:** JSON {pairs:[(chain_i, chain_j), ...]}; trains complex-level interaction topology. |
| M2 | TOP_CHAINPAIR_V1 (**TOP**) | **Q:** Select the strongest interacting chain pair (argmax under a contact-strength proxy). **Target:** JSON with top pair; bridges pairwise geometry to global complex summarization. |
| M3 | INTERFACE_TOPK_V1 (**INTF**) | **Q:** For a specified chain pair, return top-$k$ interface residues per chain. **Target:** JSON lists of residue indices; trains interface localization with explicit indexing. |
| M3 | HOTSPOT_TOPK_V1 (**HOT**) | **Q:** For a specified chain pair, return top-$k$ hotspot residues per chain. **Target:** JSON lists; pushes toward energetic salience beyond generic proximity. |

*Table 9.* **Stage I multimodal alignment ablation (schema completion).** We ablate input modalities to $E_{und}$: ✓ indicates the modality is enabled and ✗ indicates it is removed. **Struct** denotes AF3-style structure tokens from CDR-masked refolding and **Seq** denotes ESM-2 sequence embeddings. All values are accuracies measured after 10k training steps.

| Struct | Seq | num_chains | length_bin | major_ss | ss_fraction | longest_run | segments | overall |
|---|---|---|---|---|---|---|---|---|
| ✓ | ✗ | 90.9% | 65.7% | 82.8% | 53.0% | 59.8% | 78.2% | 71.7% |
| ✗ | ✓ | 88.9% | 88.2% | 88.1% | 65.6% | 67.2% | 83.7% | 80.3% |
| ✓ | ✓ | **91.8%** | **91.7%** | **86.7%** | **67.5%** | **67.8%** | **83.1%** | **81.4%** |

## C.3. Stage II: Mid-Training

**Task overview.** Stage II mid-training strengthens $E_{und}$'s structural reasoning and index-grounded representations through a curriculum of instruction-following meta-tasks with verifiable JSON outputs. Building on Stage I's chain-aware abstraction, Stage II shifts toward *residue-level grounding* and *geometry- and interface-aware reasoning* using labels deterministically derived from atomic structures. We organize Stage II into four *phases* (M0–M3), progressing from local residue attributes to pairwise geometry and, finally, complex-level interaction and interface localization.

**Structural reasoning meta-tasks across phases.** Table 10 summarizes the meta-tasks used in Stage II and their associated prompt/target formats. Phase M0 focuses on explicit residue grounding and local structural state, including residue identity retrieval (**RR**) and windowed per-residue annotations (**DSSP**, **RSA**) that encourage the model to maintain consistent (chain, position) addressing. Phase M1 introduces pairwise geometry with distance bin prediction (**DIST**) and contact classification (**CONTACT**), together with batched pair queries (**BATCH**) to enforce scalable, consistent indexing across many residue pairs. Phase M2 expands from local geometry to chemistry- and topology-aware reasoning, including salt-bridge prediction (**SALT**) and complex-level interaction structure via interacting chain-pair listing (**CHAIN**) and top interacting pair selection (**TOP**). Finally, Phase M3 targets interface understanding with explicit top-$k$ localization objectives for interface residues (**INTF**) and energetic hotspots (**HOT**) for a specified chain pair.

**Curriculum composition and replay.** Across phases, we use a curriculum that introduces new task families while maintaining a low-rate replay of earlier tasks to stabilize residue grounding and mitigate catastrophic forgetting. Table 11 reports the sampling composition for each phase: M0 is dominated by local residue annotation and identity retrieval, M1 shifts to pairwise geometry (**DIST/CONTACT**) with a small fraction of batched queries (**BATCH**), and M2 increases the emphasis on batched pair reasoning while incorporating chemistry- and topology-aware objectives (**SALT**, **CHAIN**, **TOP**). In M3, the active set expands to include explicit interface localization (**INTF**, **HOT**) and complex-level chain interaction tasks (**CHAIN**, **TOP**), while continuing to train on pairwise geometry (**DIST/CONTACT/BATCH**). Throughout M1–M3, we replay earlier local supervision (**DSSP/RSA**) at a small fixed rate to preserve per-residue structural priors.

*Table 11.* **Curriculum phases and task composition for Stage II mid-training.** Each phase mixes newly introduced reasoning tasks with low-rate replay of earlier tasks to preserve residue grounding and prevent catastrophic forgetting. Percentages indicate the relative sampling frequency of each task type. Abbreviations follow Table 10.

| Phase | Total | Active Set | Replay |
|---|---|---|---|
| M0 | 2M | **RR** (40%), **DSSP** (35%), **RSA** (25%) | None |
| M1 | 3M | **DIST** (45%), **CONTACT** (40%), **BATCH** (5%) | **DSSP** (5%), **RSA** (5%) |
| M2 | 2.64M | **DIST** (55%), **BATCH** (20%), **CONTACT** (17%) | **DSSP** (4%), **RSA** (4%) |
| M3 | 1.45M | **CHAIN** (10%), **TOP** (10%), **INTF** (10%), **HOT** (10%), **SALT** (10%) **DIST** (16%), **CONTACT** (10%), **BATCH** (18%) | **DSSP** (3%), **RSA** (3%) |

**Training results.** Table 11 defines a curriculum that progressively introduces more challenging reasoning objectives (M0→M3) while maintaining a low-rate replay of earlier tasks. The accuracy trajectories in Table 12 are consistent with the intended effect of this design: earlier capabilities are largely preserved as new tasks are introduced. For example, **DSSP** accuracy remains stable and improves by the final checkpoint ($52.0 \rightarrow 52.5 \rightarrow 51.0 \rightarrow 55.5$), and **RSA** increases substantially once later-phase training begins ($50.3 \rightarrow 52.8 \rightarrow 61.1 \rightarrow 59.4$). This pattern suggests that replay mitigates catastrophic forgetting: despite intermediate fluctuations (e.g., the modest **DSSP** dip at M2), the final M3 model retains and even improves foundational residue-level competencies relative to the M0 baseline. In parallel, the introduction of pairwise objectives yields predictable gains on geometric reasoning tasks: **DIST** improves from 27.1% to 29.2% to 39.6% as training continues, while batched pair queries rapidly become tractable ($0.0 \rightarrow 71.4 \rightarrow 84.8 \rightarrow 83.1$), indicating that the model learns both the underlying geometry and the strict, verifiable output format required for multi-query consistency. Overall, the curriculum-plus-replay strategy expands reasoning capacity across task families while maintaining performance on foundational residue-level labels.

*Table 12.* **Average task accuracy across Stage II structural reasoning tasks.** Results are averaged over eight responses per query. Reported values are percentages. Training steps correspond to the number of optimization steps at the end of each curriculum phase.

| Phase | Steps | M0 | | M1–M2 | | | M3 | | | | |
|---|---|---|---|---|---|---|---|---|---|---|---|
| | | DSSP | RSA | DIST | CONTACT | BATCH | CHAIN | HOT | INTF | SALT | TOP |
| M0 | 4,500 | 52.0% | 50.3% | 0.0% | 0.0% | 0.0% | 0.0% | 0.0% | 0.0% | 0.0% | 0.0% |
| M1 | 8,000 | 52.5% | 52.8% | 27.1% | 52.1% | 71.4% | 0.0% | 0.0% | 0.0% | 0.0% | 0.0% |
| M2 | 5,000 | 51.0% | 61.1% | 29.2% | 57.9% | 84.8% | 0.0% | 0.0% | 0.0% | 0.0% | 0.0% |
| **M3** | **3,000** | **55.5%** | **59.4%** | **39.6%** | **58.3%** | **83.1%** | **90.6%** | **11.6%** | **15.1%** | **46.9%** | **100.0%** |

## C.4. Stage III: Joint Reasoning-Guided Design

**Task overview.** Stage III couples the understanding expert ($E_{\mathrm{und}}$) with the diffusion-based generation expert ($E_{\mathrm{gen}}$) and optimizes them jointly for antibody design. The central objective is to ensure that $E_{\mathrm{und}}$ produces residue-level design signals that are not only plausible, but *causally useful* for $E_{\mathrm{gen}}$'s structure-sequence co-design. We instantiate this stage as supervised antibody CDR redesign on antibody-antigen complexes, where framework residues are fixed, and the six CDR loops are treated as designable regions. Optionally, redesign is conditioned on antigen hotspot residues specified by explicit (chain, position) indices, encouraging $E_{\mathrm{und}}$ to ground its reasoning to interface-relevant sites.

**Supervised redesign task and output format.** Table 13 defines the Stage III supervision: given an antibody-antigen complex with masked CDRs, the model must output per-loop CDR sequences in a structured JSON format. This format supports deterministic evaluation (exact parsing of loop boundaries and sequences) and, when enabled, allows auxiliary fields such as hotspot conditioning metadata and brief residue-level rationale. In contrast to Stage I/II, where supervision

targets are fully deterministic functions of structure, Stage III uses redesign targets derived from native CDR sequences (and optional hotspot annotations) to directly train the model for downstream generative utility.

*Table 13.* **Stage III supervised antibody CDR redesign task for joint reasoning-guided design.** The task requires redesigning masked antibody CDRs under a fixed framework, optionally conditioned on antigen hotspot residues. Targets are structured JSON outputs specifying per-CDR sequences and optional residue-level rationale, enabling explicit reasoning supervision and deterministic evaluation.

| Stage | Task Type | Question / Target Formats |
|---|---|---|
| D0 | AB_CDR_REDESIGN_SFT_V1 | **Q:** Redesign antibody CDR sequences under a fixed framework, optionally conditioned on antigen hotspots.
**Target:** JSON specifying per-loop CDR sequences, with optional rationale or conditioning fields. |

**Evaluation protocol and CDR metrics.** We evaluate supervised CDR redesign using a three-level protocol that captures complementary failure modes: (i) **CDR detection**, which measures whether the model identifies the correct CDR locations (boundaries) along the antibody chains; (ii) **sequence-level correctness**, which measures similarity between the predicted and reference CDR sequences as whole strings under substitutions and indels; and (iii) **residue-level correctness**, which measures per-position agreement after alignment. Concretely, given predicted CDR residue sets (or regions) and predicted CDR sequences, we report:

- **CDR detection metrics:** recall, precision, and set match by treating each CDR as a set of within-chain residue indices and comparing the predicted set to the ground-truth set; set match is strict and requires an exact boundary match.

- **Sequence-level metrics:** string similarity between the predicted CDR sequence $\hat{s}$ and the ground-truth sequence $s$, including Edit Similarity and normalized LCS, where Edit Similarity is computed from the Levenshtein edit distance $d_{\text{edit}}(\hat{s}, s)$ as $\text{EditSim}(\hat{s}, s) = 1 - \frac{d_{\text{edit}}(\hat{s}, s)}{\max(|\hat{s}|, |s|)}$, and normalized longest common subsequence is $\text{LCS\_norm}(\hat{s}, s) = \frac{\text{LCS}(\hat{s}, s)}{|s|}$.

- **Residue-level metrics:** per-residue agreement after sequence alignment (using the same alignment procedure for all methods), including identity-based criteria (e.g., position accuracy and set-based precision/recall/F1) and a substitution-matrix score that credits conservative substitutions. Specifically, BLOSUM62 is computed by scoring each aligned residue–residue pair $(\hat{a}_t, a_t)$ with the BLOSUM62 entry $B(\hat{a}_t, a_t)$ and reporting the mean over aligned, non-gap positions: $\text{BLOSUM62} = \frac{1}{T} \sum_{t=1}^{T} B(\hat{a}_t, a_t)$, where $T$ is the number of aligned residue–residue positions.

Together, these metrics provide a complete view of redesign quality: detection captures localization, sequence-level metrics capture global string similarity under indels, and residue-level metrics capture site-wise recovery and biochemical plausibility.

**Training results and CDR redesign ablation.** Tables 14 and 15 show that enabling both **Thinking** (CoT-style reasoning supervision applied to $\mathbf{E}_{\text{und}}$) and **Playback** (low-rate replay of earlier Stage II tasks during Stage III training) yields the most consistent improvements for supervised CDR redesign. In Table 14, the combined setting achieves the strongest overall performance across detection and sequence-level criteria, including the best CDR detection recall/precision/set match (99.70/98.90/95.77) and the highest example- and sequence-level scores (e.g., EMR 1.08; length match 58.76). These gains are mirrored in residue-level metrics, where the combined setting attains the best or near-best values (Pos Acc 33.85; F1 32.90; BLOSUM62 54.91). This pattern is consistent with the role of replay observed in Stage II: **Playback** stabilizes previously learned structural and formatting competencies during later optimization, improving robustness when the model must generate strict, structured outputs under masked constraints. Meanwhile, **Thinking** most strongly benefits example-level and sequence-level measures (e.g., EMR and length match), which are particularly sensitive to global coherence and constraint satisfaction rather than purely token-wise correctness.

The per-CDR breakdown in Table 15 further indicates that improvements are uneven across loops, consistent with known difficulty differences among CDRs. H1 and L2 show the largest relative gains in EMR and position accuracy under the combined setting, whereas H3 remains the most challenging region (near-zero EMR across all settings and low position accuracy), reflecting its higher structural variability and sequence diversity. Notably, residue-level metrics across ablations are comparatively close, suggesting partial saturation and higher sensitivity to evaluation variance at the token level; in

contrast, the consistent gains in example-level and sequence-level metrics provide a clearer signal of net improvement. Overall, these results support the conclusion that replay is beneficial for end-to-end antibody redesign, and that combining replay with explicit reasoning supervision yields the strongest performance under both CDR detection and sequence generation criteria.

*Table 14.* **Ablation study on antibody CDR evaluation metrics across detection, sequence-level, and residue-level criteria.** Detection: Recall (↑), Precision (↑), Set Match (↑). Sequence-level: EMR (↑), Edit Sim (↑), LCS (↑), Length Match (↑). Residue-level: Pos Acc (↑), Precision (↑), Recall (↑), F1 (↑), BLOSUM62 (↑).

| Playback | Thinking | CDR Detection | | | Sequence-Level Metrics | | | | Residue-Level Metrics | | | | |
|---|---|---|---|---|---|---|---|---|---|---|---|---|---|
| | | Recall | Precision | Set Match | EMR | Edit Sim | LCS | Length Match | Pos Acc | Precision | Recall | F1 | BLOSUM62 |
| ✗ | ✗ | 99.55% | 97.95% | 93.12% | 0.59% | **37.94%** | **44.02%** | 43.80% | 31.11% | 30.23% | 29.85% | 29.88% | 52.97% |
| ✗ | ✓ | 99.63% | **98.94%** | 95.24% | 1.03% | 36.97% | 43.81% | 57.19% | 33.50% | 32.73% | 32.61% | 32.55% | 54.76% |
| ✓ | ✗ | 99.46% | 98.17% | 93.12% | 0.54% | 36.94% | 43.03% | 40.55% | 30.21% | 29.31% | 28.75% | 28.85% | 52.13% |
| ✓ | ✓ | **99.70%** | 98.90% | **95.77%** | **1.08%** | 37.28% | 43.90% | **58.76%** | **33.85%** | **33.09%** | **32.95%** | **32.90%** | 54.91% |

*Table 15.* **Per-CDR performance metrics.** EMR: Exact Match Rate (↑), Pos: Position Accuracy (↑), Edit: Edit Similarity (↑).

| Playback | Thinking | H1 | | | H2 | | | H3 | | | L1 | | | L2 | | | L3 | | |
|---|---|---|---|---|---|---|---|---|---|---|---|---|---|---|---|---|---|---|---|
| | | EMR | Pos | Edit | EMR | Pos | Edit | EMR | Pos | Edit | EMR | Pos | Edit | EMR | Pos | Edit | EMR | Pos | Edit |
| ✗ | ✗ | 2.1% | 52.6% | 53.5% | 0.0% | 23.7% | **30.2%** | 0.0% | **11.7%** | **24.0%** | 0.7% | 39.4% | **46.7%** | 0.7% | 30.5% | 36.0% | 0.0% | 30.2% | **38.7%** |
| ✗ | ✓ | **3.4%** | **54.1%** | **54.6%** | 0.0% | **25.9%** | 29.2% | 0.0% | 11.1% | 20.7% | 1.0% | 38.5% | 43.2% | 1.7% | 38.4% | 38.5% | 0.0% | **35.4%** | 37.1% |
| ✓ | ✗ | 2.7% | 50.8% | 51.9% | 0.0% | 22.2% | 29.5% | 0.0% | 11.0% | 22.8% | 0.0% | 38.3% | 44.8% | 0.3% | 29.9% | 36.7% | 0.0% | 30.6% | 37.6% |
| ✓ | ✓ | 3.2% | 52.4% | 52.6% | **0.0%** | 25.0% | 29.2% | **0.0%** | 10.7% | 20.7% | **1.3%** | **41.2%** | 44.1% | **2.0%** | **42.0%** | **42.0%** | **0.0%** | 35.3% | 37.4% |

# D. Schema Examples for Training Supervision

Schema-style supervision provides an expandable, verifiable, machine-readable, and easily maintained framework for expressing verifiable training targets. Because schema separates *what* must be grounded from *how* it is described linguistically, the same interface can be reused across heterogeneous data modalities. In this work, we apply the schema framework to PDB-derived protein structures and antibody–antigen complexes, enabling unambiguous chain- and position-resolved supervision with low-variance targets. Compared with free-form text, schemas reduce linguistic redundancy and constrain outputs to standardized fields, improving controllability and auditability. Importantly, schema outputs can be losslessly rendered into natural language using an LLM-based agent, thereby preserving readability and captioning-style behaviors when needed. Looking forward, the same schema abstraction can be extended beyond structural data to systematically extract and normalize information from scientific literature.

## D.1. Stage I Schema Examples

**Stage I: Alignment Schema Example (*Single-chain*)**

Here we show a representative Stage I alignment supervision example for a single-chain protein. The target is a compact, machine-readable schema that summarizes global and chain-level structural properties (e.g., chain count, length bin, and secondary-structure statistics). All fields are computed from the same underlying structure and are directly verifiable against the PDB-derived features, providing low-variance supervision for projector-level multi-modal alignment.

**Input structure.** PDB `8AF7`, chain A (processed biological assembly).

**Schema query and target.**

**Schema completion (alignment)**
```
<TASK=ALIGNMENT_SCHEMA_B1_V2>
```
Return the schema for this structure in JSON format.

*Output:*

```json
{
  "task_type": "ALIGNMENT_SCHEMA_B1_V2",
  "global": {
    "num_chains": 1
  },
  "chains": [
    {
      "chain_id": "A",
      "length_bin": "200-300",
      "secondary_structure_fraction_bins": {
        "H": "40-50",
        "E": "20-30",
        "C": "20-30"
      },
      "major_secondary_structure": "H",
      "secondary_structure_longest_run_bins": {
        "H": "9-15",
        "E": "16-30",
        "C": "5-8"
      },
      "secondary_structure_segment_count_bins": {
        "H": ">5",
        "E": ">5"
      }
    }
  ]
}
```

**Interpretation.** This schema provides a standardized, verifiable structural summary that the model must reconstruct

from structure-conditioned tokens (rather than from sequence text). The use of coarse bins reduces linguistic and numeric variance while preserving salient global shape cues, thereby anchoring the learned projection in a consistent schema space before introducing additional relational objectives in Stage II.

## Stage I: Alignment Schema Example (*Multiple-chain*)

We provide a second Stage I alignment datapoint of the same schema type, illustrating a multi-chain assembly. The target uses the identical field set as the single-chain example, but repeats the per-chain descriptor block for each chain ID. As in the single-chain case, all fields are computed from the underlying structure and are directly verifiable from PDB-derived annotations, enabling consistent supervision across heterogeneous complexes (H=helix, E=strand, C=coil, NA=missing).

**Input structure.** PDB `7ATF`, chains A and B (processed biological assembly).

**Schema query and target.**

**Schema completion (alignment)**
`<TASK=ALIGNMENT_SCHEMA_B1_V2>`
Return the schema for this structure in JSON format.

*Output:*

```
{
  "task_type": "ALIGNMENT_SCHEMA_B1_V2",
  "global": {
    "num_chains": 2
  },
  "chains": [
    {
      "chain_id": "A",
      "length_bin": "200-300",
      "secondary_structure_fraction_bins": {
        "H": "50-60",
        "E": "20-30",
        "C": "20-30"
      },
      "major_secondary_structure": "H",
      "secondary_structure_longest_run_bins": {
        "H": "16-30",
        "E": "9-15",
        "C": "9-15"
      },
      "secondary_structure_segment_count_bins": {
        "H": ">5",
        "E": ">5"
      }
    },
    {
      "chain_id": "B",
      "length_bin": "200-300",
      "secondary_structure_fraction_bins": {
        "H": "50-60",
        "E": "20-30",
        "C": "20-30"
      },
      "major_secondary_structure": "H",
      "secondary_structure_longest_run_bins": {
        "H": "16-30",
        "E": "9-15",
        "C": "9-15"
      },
      "secondary_structure_segment_count_bins": {
```

```
            "H": ">5",
            "E": ">5"
         }
       }
    ]
  }
```

**Interpretation.** This two-chain example illustrates how the alignment schema generalizes to assemblies by instantiating the same per-chain descriptor block for each chain ID. The resulting output remains compact, machine-readable, and directly verifiable from PDB-derived features, while enforcing explicit chain disambiguation. This uniform schema, therefore, supports consistent multi-modal alignment across proteins with varying chain counts and secondary-structure profiles.

---

### Stage I: Multi-modal Alignment (*Caption-style*)

We show a representative Stage I alignment supervision example using a caption-style target (ALIGNMENT_CAPTION_B1_V2). The model is prompted to produce a short natural-language summary of the same low-entropy, binned structural descriptors used in schema completion (e.g., chain count, length bin, and secondary-structure statistics), preserving readability while remaining directly verifiable from PDB-derived features.

**Input structure.** PDB 9P9U, chain A (processed biological assembly).

**Schema query and target.**

> **Caption completion (alignment)**
> <TASK=ALIGNMENT_CAPTION_B1_V2>
> Return the alignment caption for this structure.
>
> *Output:*
>
> ```
> This structure has a single chain (Chain A), about 500-800 residues long. Its
> secondary structure is helix-dominant, with roughly 40-50% helices (H),
> 20-30% beta strands (E), and 20-30% coils (C). The longest continuous
> helix stretch is longer than 30 residues, while the longest strand stretch
> is about 16-30 residues and the longest coil stretch is about 9-15 residues.
> ```

**Interpretation.** Compared with strict JSON schemas, caption targets provide a human-readable supervision signal that can improve instruction-following and narrative grounding, while still constraining outputs to a fixed set of binned, structure-derived fields. We use tight templates to minimize linguistic redundancy and semantic leakage, ensuring that correct captions must be supported by the injected structure/sequence tokens rather than textual priors.

---

## D.2. Stage II: Mid-Training

### Stage II Mid-Training Task Bundle (*Multiple-chain*)

We illustrate a compact bundle of Stage II mid-training queries for a single structure. For each task type, we select one representative instance (deterministically) and present the corresponding query and target. All residue indices are **1-based within-chain**.

**Input structure.** PDB 8JRK (multi-chain complex; canonical ordering). The interacting chain pair in this example is C--D. (processed biological assembly)

**Selected queries and targets (one per task type).**

> **(a) Residue identity grounding**
> <TASK=RESIDUE_RETRIEVAL_V1>
> What is the amino-acid type at residue (chain D, position 22)? Choose one option:

{A,C,D,E,F,G,H,I,K,L,M,N,P,Q,R,S,T,V,W,Y,X}.

*Output:*

```
{"aa": "E"}
```

**(b) Secondary-structure labeling over a window (H=helix, E=strand, C=coil, NA=missing)**
<TASK=DSSP_SEQ_V1>
What are the secondary-structure labels for chain B residues 48 through 52 (inclusive)? Output a 5-character string using {H,E,C,NA}.

*Output:*

```
{"labels": "EECCH"}
```

**(c) Solvent-accessibility profiling (B=buried, M=mid, E=exposed, NA=missing)**
<TASK=RSA_SEQ_V1>
What are the solvent-accessibility bins for chain E residues 1 through 5 (inclusive)? Output a 5-character string using {B,M,E,NA}.

*Output:*

```
{"labels": "EEBEM"}
```

**(d) Pairwise contact (binary)**
<TASK=PAIR_CONTACT_YN_V1>
Are residue (chain C, position 74) and residue (chain D, position 21) in contact under the $< 8.0\text{Å}$ C$\beta$ (C$\alpha$ for Gly) rule? Choose one option: {Contact, NotContact}.

*Output:*

```
{"choice": "Contact"}
```

**(e) Pairwise distance bin (class-specific bins)**
<TASK=PAIR_DIST_BIN_V1>
What is the distance bin between residue (chain C, position 74) and residue (chain D, position 21)? Pair class: cross_chain. Choose one option: {<4, 4-6, 6-8, 8-10, 10-14, >14}.

*Output:*

```
{"dist_bin": "4-6"}
```

**(f) Batch distance prediction (first 5 of 30 shown)**
<TASK=PAIR_BATCH_V1>
For each residue pair below, what is the distance bin (using the bin set for its pair class)? Return JSON only.

*Input (truncated):*

```
{"pairs": [
  {"pair_id":"p1","i":{"chain":"C","pos":44},"j":{"chain":"C","pos":45}},
  {"pair_id":"p2","i":{"chain":"B","pos":21},"j":{"chain":"B","pos":24}},
  {"pair_id":"p3","i":{"chain":"C","pos":4},"j":{"chain":"C","pos":10}},
  {"pair_id":"p4","i":{"chain":"A","pos":77},"j":{"chain":"A","pos":78}},
  {"pair_id":"p5","i":{"chain":"B","pos":151},"j":{"chain":"B","pos":156}}
]}
```

*Output (truncated):*

```
{"pairs": [
  {"pair_id":"p1","dist_bin":"<3.5"},
  {"pair_id":"p2","dist_bin":"<3.5"},
  {"pair_id":"p3","dist_bin":"<3.5"},
  {"pair_id":"p4","dist_bin":"<3.5"},
  {"pair_id":"p5","dist_bin":"3.5-5"}
]}
```

*(Note: the full training record contains 30 pairs; we truncate for readability.)*

**(g) Chain-pair interaction graph**
<TASK=CHAINPAIR_GRAPH_V1>
Which chain pairs are interacting in this complex under the strength threshold used by the dataset generator? Return JSON only.

*Output:*

```
{"pairs": [{"chain_i": "C", "chain_j": "D"}]}
```

**(h) Top interacting chain pair**
<TASK=TOP_CHAINPAIR_V1>
Which chain pair has the strongest interaction in this complex under the dataset generator's contact-strength rule? Return JSON only.

*Output:*

```
{"top_chain_pair": {"chain_i": "C", "chain_j": "D"}}
```

**(i) Interface localization (top-$K$ residues)**
<TASK=INTERFACE_TOPK_V1>
For the chain pair (C,D), which residues form the top-10 interface on each chain under the dataset generator's interface rule? Return JSON only.

*Output:*

```
{"chain_pair":{"chain_i":"C","chain_j":"D"},"topk":10,
 "C":[73,52,9,5,82,7,96,69,51,143],
 "D":[6,7,9,8,11,20,89,33,151,13]}
```

**(j) Hotspot localization (top-$K$ residues)**
<TASK=HOTSPOT_TOPK_V1>
For the chain pair (C,D), which residues are the top-5 hotspots on each chain under the dataset generator's hotspot proxy rule? Return JSON only.

*Output:*

```
{"chain_pair":{"chain_i":"C","chain_j":"D"},"topk":5,
 "C":[73,52,82,7,9],
 "D":[6,7,8,9,11]}
```

**(k) Salt-bridge bin (top chain pair)**
<TASK=SALTBRIDGE_BIN_V1>
For the chain pair (C,D), what is the salt-bridge count bin under the dataset generator's salt-bridge rule? Choose one option: {0, 1-2, 3-5, 6-10, 11-20, >20}.

*Output:*

```
{"salt_bridge_bin": "6-10"}
```

**Interpretation.** Showing one instance per task type makes Stage II supervision easy to audit and mechanically verifiable. Local labeling tasks (residue identity, DSSP, RSA) encourage precise token-level grounding, pairwise contact/distance tasks enforce geometric consistency, and chain/interface tasks encourage multi-chain interaction reasoning and localization. For long structured prompts (e.g., PAIR_BATCH_V1), we truncate the displayed JSON for readability while keeping the underlying training record unchanged.

## Stage II Mid-Training Task Bundle (*Single-chain*)

We show a representative Stage II mid-training record for a single-chain protein, including one illustrative instance per applicable task type. All residue indices are **1-based within-chain**. For long structured queries (e.g., PAIR_BATCH_V1), we display only the first pair and truncate the remainder for readability, while leaving the underlying training record unchanged.

**Input structure.** PDB 6SW1, chain A (processed biological assembly).

**Schema queries and targets.**

### (a) Residue identity grounding
<TASK=RESIDUE_RETRIEVAL_V1>
What is the amino-acid type at residue (chain A, position 143)? Choose one option:
{A,C,D,E,F,G,H,I,K,L,M,N,P,Q,R,S,T,V,W,Y,X}.

*Output:*

{"aa":"Q"}

### (b) Secondary-structure labeling over a window (H=helix, E=strand, C=coil, NA=missing)
<TASK=DSSP_SEQ_V1>
What are the secondary-structure labels for chain A residues 277 through 281 (inclusive)? Output a 5-character string using {H,E,C,NA}.

*Output:*

{"labels":"EECCC"}

### (c) Solvent-accessibility profiling (B=buried, M=mid, E=exposed, NA=missing)
<TASK=RSA_SEQ_V1>
What are the solvent-accessibility bins for chain A residues 195 through 199 (inclusive)? Output a 5-character string using {B,M,E,NA}.

*Output:*

{"labels":"MMEMB"}

### (d) Pairwise contact classification
<TASK=PAIR_CONTACT_YN_V1>
Are residue (chain A, position 58) and residue (chain A, position 201) in contact under the $< 8.0\mathring{A}$ C$\beta$ (C$\alpha$ for Gly) rule? Choose one option: {Contact, NotContact}.

*Output:*

{"choice":"NotContact"}

### (e) Pairwise distance-bin prediction
<TASK=PAIR_DIST_BIN_V1>
What is the distance bin between residue (chain A, position 70) and residue (chain A, position 223)? Pair class: long. Choose one option: { $<$6, 6-8, 8-10, 10-12, 12-16, $>$16}.

*Output:*

```
{"dist_bin":">16"}
```

**(f) Batch distance-bin prediction (truncated)**
`<TASK=PAIR_BATCH_V1>`
For each residue pair below, what is the distance bin (using the bin set for its pair class)? Return JSON only.
*Input (first pair shown; remaining pairs truncated):*

```
{"pairs":[{"pair_id":"p1","i":{"chain":"A","pos":77},
"j":{"chain":"A","pos":82}}, ... ]}
```

*Output (first result shown; remaining results truncated):*

```
{"pairs":[{"pair_id":"p1","dist_bin":"<3.5"}, ... ]}
```

**Interpretation.** In the single-chain setting, Stage II supervision emphasizes residue-level grounding (residue identity, local secondary structure, and solvent exposure) together with intra-chain geometric reasoning (contact classification and short/long-range distance bins). Cross-chain interaction tasks are not instantiated because no inter-chain interfaces exist.

---

## Stage II Mid-Training Task Bundle (*Antibody complex*)

Because our downstream objective is antibody redesign, Stage II mid-training includes structure-grounded supervision on antibody-antigen complexes in addition to single proteins. Here we show a representative mid-training record derived from a single antibody complex, with tasks that jointly cover residue-level grounding, coarse structural patterns, geometry, multi-chain interaction topology, interface/hotspot localization, and CDR-level quality. All fields are computed from the same underlying complex and are directly verifiable against PDB-derived features.

**Input structure.** `7ran_E_e_BC` (processed biological assembly), antibody chains `B,C` and antigen chain `E`.

**Schema queries and targets.**

**(a) CDR quality prediction (LDDT bin)**
`<TASK=LDDT_BIN_V1>`
Predict the CDR LDDT score bin for this antibody structure. Choose one option: {<0.5, 0.5-0.6, 0.6-0.7, 0.7-0.8, >0.8}.
*Output:*

```
{"lddt_bin":"0.7-0.8"}
```

**(b) Residue identity grounding**
`<TASK=RESIDUE_RETRIEVAL_V1>`
What is the amino-acid type at residue (chain C, position 25)? Choose one option:
{A,C,D,E,F,G,H,I,K,L,M,N,P,Q,R,S,T,V,W,Y,X}.
*Output:*

```
{"aa":"C"}
```

**(c) Secondary-structure labeling over a window (H=helix, E=strand, C=coil, NA=missing)**
`<TASK=DSSP_SEQ_V1>`
What are the secondary-structure labels for chain B residues 29 through 33 (inclusive)? Output a 5-character string using {H,E,C,NA}.

*Output:*

```
{"labels":"HHHCE"}
```

**(d) Solvent-accessibility profiling (B=buried, M=mid, E=exposed, NA=missing)**
<TASK=RSA_SEQ_V1>
What are the solvent-accessibility bins for chain B residues 99 through 103 (inclusive)? Output a 5-character string using {B,M,E,NA}.

*Output:*

```
{"labels":"BBMEB"}
```

**(e) Pairwise contact classification (cross-chain)**
<TASK=PAIR_CONTACT_YN_V1>
Are residue (chain B, position 105) and residue (chain C, position 11) in contact under the $< 8.0\mathring{A}$ C$\beta$ (C$\alpha$ for Gly) rule? Choose one option: {Contact, NotContact}.

*Output:*

```
{"choice":"NotContact"}
```

**(f) Pairwise distance bin (cross-chain)**
<TASK=PAIR_DIST_BIN_V1>
What is the distance bin between residue (chain B, position 105) and residue (chain C, position 11)? Pair class: cross_chain. Choose one option: {<4, 4-6, 6-8, 8-10, 10-14, >14}.

*Output:*

```
{"dist_bin":">14"}
```

**(g) Batch distance-bin prediction (truncated)**
<TASK=PAIR_BATCH_V1>
For each residue pair below, what is the distance bin (using the bin set for its pair class)? Return JSON only.
*Input (first pair shown; remaining pairs truncated):*

```
{"pairs":[{"pair_id":"p1","i":{"chain":"C","pos":308},
"j":{"chain":"C","pos":309}}, ... ]}
```

*Output (first result shown; remaining results truncated):*

```
{"pairs":[{"pair_id":"p1","dist_bin":"<3.5"}, ... ]}
```

**(h) Chain-pair interaction graph**
<TASK=CHAINPAIR_GRAPH_V1>
Which chain pairs are interacting in this complex under the strength threshold used by the dataset generator? Return JSON only.

*Output:*

```
{"pairs":[{"chain_i":"C","chain_j":"E"}]}
```

**(i) Top interacting chain pair**
<TASK=TOP_CHAINPAIR_V1>
Which chain pair has the strongest interaction in this complex under the dataset generator's contact-strength rule? Return JSON only.

*Output:*

```
{"top_chain_pair":{"chain_i":"C","chain_j":"E"}}
```

**(j) Antibody-antigen interface localization (Top-10)**
`<TASK=INTERFACE_TOPK_V1>`
For the antibody-antigen chain pair (C,E), which residues form the top-10 interface on each chain under the dataset generator's interface rule? Return JSON only.

*Output:*

```
{"chain_pair":{"chain_i":"C","chain_j":"E"},"topk":10,
 "C":[283,281,284,279,282,324,280,40,298,300],
 "E":[25,23,24,1,4,26,8,21,28,22]}
```

**(k) Antibody-antigen hotspot localization (Top-5)**
`<TASK=HOTSPOT_TOPK_V1>`
For the antibody-antigen chain pair (C,E), which residues are the top-5 hotspots on each chain under the dataset generator's hotspot proxy rule? Return JSON only.
*Output:*

```
{"chain_pair":{"chain_i":"C","chain_j":"E"},"topk":5,
 "C":[283,281,279,284,40],
 "E":[25,23,24,1,4]}
```

**(l) Salt-bridge bin over the top chain pair**
`<TASK=SALTBRIDGE_BIN_V1>`
For the chain pair (C,E), what is the salt-bridge count bin under the dataset generator's salt-bridge rule? Choose one option: {0, 1-2, 3-5, 6-10, 11-20, >20}.

*Output:*

```
{"salt_bridge_bin":"1-2"}
```

**Interpretation.** In the antibody-complex setting, Stage II supervision extends residue-level grounding and pairwise geometric reasoning with interaction-aware objectives that are directly aligned with downstream design. Interface and hotspot localization provide explicit, verifiable signals for where binding is mediated, while the CDR LDDT bin adds a coarse structural-quality target that encourages plausible loop conformations alongside accurate antibody-antigen geometry.

## D.3. Stage III: Joint Reasoning-Guided Design

### Stage III: Joint Reasoning-Guided Design

We illustrate a single Stage III training datapoint for antibody CDR redesign (`AB_CDR_REDESIGN_SFT_V1`). The prompt specifies a small set of antigen design points (hotspot residues) and asks the understanding expert to return a structured JSON object that (i) records the hotspot-aware reasoning trace and (ii) enumerates redesigned CDR sequences in an unambiguous, residue-level format.

**Input complex.** PDB `7SBZ`; antibody heavy chain A, light chain B; antigen chain D (processed complex).

**Schema query and target.**

**CDR redesign with hotspot-conditioned schema output**
`<TASK=AB_CDR_REDESIGN_SFT_V1>`
You are redesigning masked CDR regions of an antibody to improve binding to the antigen.
Design points (antigen hotspots): `[D,179]`, `[D,237]`, `[D,208]`.
Output JSON with keys: `task`, `thinking`, `answer`.

*Output:*

```
{
  "task": "AB_CDR_REDESIGN_SFT_V1",
```

```
    "thinking": {
      "design_points": [{"ag_chain":"D","ag_pos":179}, ...]
      "hotspots_where": [{"ag_chain":"D","ag_pos":179,"atomic_contact_count":34,
                          "delta_sasa_A2":115.63,"is_hotspot":true}, ...],
      "shape_context": [{"ag_chain":"D","ag_pos":179,"rsa_label":"Exposed"}, ...],
      "chemistry_logic": [{"ag_chain":"D","ag_pos":179,"ag_res":"K","ag_charge":1,
                          "interaction_types":["Van der Waals"]}, ...],
      "binder_solution": [{"ag_chain":"D","ag_pos":179,
                          "binder_contacts":[{"ab_chain":"A",
                                "ab_pos":123,"cdr":"H3"}, ...]
    },
    "answer": {
      "cdrs_present": ["H1","H2","H3","L1","L2","L3"],
      "cdr_sequences": {
        "H1": {"len": 7,  "seq": "<HCDR1>XYTXXSY</HCDR1>",
               "filled_positions":[{"pos":2,"aa":"Y"},{"pos":3,"aa":"T"}, ...]},
        "H3": {"len": 13, "seq": "<HCDR3>XLKYDGSTYRXXX</HCDR3>",
               "filled_positions":[{"pos":2,"aa":"L"},{"pos":3,"aa":"K"}, ...]},
        "L1": {"len": 16, "seq": "<LCDR1>XXXXXXXXXNXXXYXX</LCDR1>",
               "filled_positions":[{"pos":10,"aa":"N"},{"pos":14,"aa":"Y"}]}, ...
      }
    }
  }
```

**Interpretation.** This schema enforces chain- and position-grounded redesign: antigen hotspots are specified as (*chain*, *position*) design points, and the model must return redesigned CDR loops as a standardized JSON object that explicitly enumerates loop presence and length together with per-position amino-acid assignments. The `thinking` fields provide verifiable intermediate signals (hotspot localization, exposure/geometry context, and contact-based binder mapping) that are injected into the diffusion expert via cross-attention during denoising. Joint training allows gradients from the diffusion objective to shape the understanding expert's residue-level design signals so they are optimized for downstream generative success rather than proxy supervision alone.

