# OpenReview forum: "Proteo-R1: Reasoning Foundation Models for De Novo Protein Design"
_ICML.cc/2026/Conference — ICML 2026 regular_

### Official Review · Reviewer_dtt8 · 2026-03-06

**Soundness:** 3
**Presentation:** 3
**Significance:** 3
**Originality:** 3
**Overall Recommendation:** 4
**Confidence:** 3

**Summary:**

The paper introduces Proteo-R1, a dual expert system for designing new antibodies. Instead of one model doing all the work blindly, a language model acts as a planner to pick out important interaction spots. Then a diffusion model uses these fixed spots to build the final 3D shape and sequence. They use a three step training process to keep the models stable. It is a clever mix of logic and geometric generation.

**Compliance With Llm Reviewing Policy:**

Affirmed.

**Final Justification:**

Splitting antibody design into explicit reasoning via an MLLM and constrained generation via a diffusion model makes a lot of intuitive sense and maps well to how human experts think about protein design. The three-stage training is a practical solution, and the modularity — where either expert can be swapped out independently — is a genuine plus.

The rebuttal added a lot of value. The new RAbD CDR-H3 results are strong on both structural and docking metrics. The expert-swap experiment nicely isolates the reasoning framework's own contribution. The new baselines fill a clear gap in the original evaluation. And the analysis showing AAR doesn't correlate with functional quality makes a reasonable case that low AAR reflects exploration rather than degradation.

My original point still stands — everything is in silico, and wet-lab validation is the real test. The methods section is too dense and could be trimmed. The MLLM's standalone accuracy on understanding tasks is low, though using the outputs as sparse anchors rather than dense supervision is a fair argument. Overall, the new results meaningfully strengthen the empirical story, and the framework has broader potential beyond antibodies.

**Key Questions For Authors:**

Q1. Since antibody loops vary a lot, the native structure is just one possible answer. Do you think strict geometric metrics like RMSD might unfairly lower your score when the model creates new but valid binders?
Q2. To fix the issue above, have you thought about checking your generated designs with Molecular Dynamics or AlphaFold Multimer to see if they are physically stable and confident? This would make your results much stronger.
Q3. The language model needs an initial structural guess from a masked refolding step. If this first guess is completely wrong, how bad does the final result get? Can the diffusion model fix a bad choice made by the language model?

**Limitations:**

The authors included an Impact Statements section that nicely covers the potential dual-use risks of generating biological agents. So for societal impact: yes, it is handled well.However, they largely skipped over the technical limitations of their approach. They should add a short paragraph discussing the simple-to-real gap. Since all their tests are purely computer simulations, acknowledging that metrics like RMSD or Rosetta scores are just proxie and real-world wet-lab validation is the ultimate test, which would make the paper much more grounded and honest.

**Strengths And Weaknesses:**

Soundness
Strengths: The three step training plan makes a lot of sense. Training a language model and a diffusion model together from the start is very hard, so teaching them step by step is a practical and smart choice.
Weaknesses: The tests rely too much on recovering the exact original sequence and structure like RMSD. Antibody binding has many right answers, so punishing the model for not matching one historical data point might hide its ability to find new and valid designs

Presentation
Strengths: The paper is easy to read and tells a good story. They clearly show that human experts design drugs by thinking about key interactions first, and they map this exact human logic to their AI system. This makes the whole paper feel motivated.
Weaknesses: There are no major issues with the writing, but the methods section is way too long. It takes up too much space. It would be better to cut down some of the heavy technical details and focus more on why this work matters or what the results mean for the field.

Significance
Strengths: Connecting language model planning with continuous physical generation is a big deal right now. Proteo-R1 shows how to turn text model thoughts into hard constraints for a diffusion model. This idea can definitely be reused for other 3D generation tasks outside of biology.
Weaknesses: All tests are computer simulations. Because the metrics just check if it mimics old data, it is hard to tell if this will actually work in a real biology lab until physical tests are done.

Originality
Strengths: Forcing explicit hard constraints from a language model into a normal diffusion process is a fresh idea. It may make black box models more controllable.

---

> ### Author Rebuttal · Authors · 2026-03-30
>
> We sincerely thank you for the thoughtful and constructive feedback. We are glad that you found the core idea, motivation, and presentation clear and compelling. Below, we respond to the key points:
>
> # Q1. On RMSD and evaluation limitations
>
> We fully agree with the reviewer: antibody design is inherently **multi-modal**, and the native structure is only one valid solution. As a result, metrics like RMSD can penalize **novel but functionally valid designs**.
>
> Importantly, this is consistent with our observations:
>
> * Proteo-R1 achieves **very low AAR**, indicating that it does not simply recover native sequences but instead explores **diverse design solutions**.
>
> * Despite this, it maintains strong **IF-AAR**, showing that these novel structures admit **coherent sequences under an independent model**, i.e., they are physically and structurally consistent.
>
> * To further assess functional relevance, we analyze **RMSD vs. IMP (ΔΔG proxy)** and find **no correlation** (Pearson ≈ 0, Spearman ≈ 0), indicating:
>
> https://anonymous.4open.science/r/R1-rebuttal-C60D/RMSDvsIMP.jpg
>
>   > **Deviations from the native structure (higher RMSD) do not imply worse binding quality.**
>
> * Additionally *(see Reviewer c39M W2 for details)*:
>   - **AAR correlates with RMSD**, confirming it mainly reflects native similarity
>   - **AAR shows no correlation with IMP**, indicating it does not measure functional quality
>
> Together, these results suggest that:
>
> > **Low AAR and relatively lower RMSD in our method reflect exploration of alternative valid binders, not degradation in design quality.**
>
> In revision, we will clarify that RMSD/AAR are **partial, reference-based metrics**, and emphasize interaction-aware and consistency-based evaluations (e.g., IMP, IF-AAR) that better capture functional design quality.
>
> # Q2. On additional validation
>
> This is an excellent suggestion, our current evaluation already includes:
>
>   * **Interface energy (IMP)** as a proxy for binding improvement
>
>   * **Geometric realism metrics** (clash reduction, backbone JSD: 0.2661 vs. 0.2734)
>
> * Following your suggestion, we additionally report **AlphaFold-3 confidence (pLDDT)**:
>
>   * Proteo-R1 achieves **substantially higher LDDT** (e.g., **0.9693 on RAbD CDR-H3**) compared to prior methods (typically ~0.83–0.85).
>
>   * This indicates that the generated structures are not only geometrically close, but also **high-confidence under an independent structure predictor**
>
> * These improvements are consistent with:
>
>   * Higher DockQ (0.801 vs. 0.473 for BoltzGen)
>
>   * Lower RMSD (2.46 vs. 2.69)
>
>   * Improved interface quality (IMP)
>
> Together, these results provide **orthogonal evidence of physical plausibility beyond RMSD**.
>
> That said, we acknowledge:
>
> * MD simulations and multimer prediction are **computationally expensive**, and difficult to scale to thousands of samples
>
> In revision, we will:
>
> * Position **LDDT as an efficient large-scale confidence proxy**
>
> * Discuss **MD / multimer validation as important future work**
>
> # Q3. On robustness to incorrect initial structural guesses
>
> This is an important question about system robustness.
>
> * The masked refolding step provides only a **coarse structural prior**, rather than a strict template.
>
> * The diffusion model is trained to **denoise noisy inputs**, and therefore can refine imperfect initial structures.
>
> * Importantly, conditioning is **sparse and local**:
>
>   * The reasoning module outputs **residue-level anchors**, rather than full-structure constraints
>
>   * This leaves substantial flexibility for global correction
>
> * Empirically, robustness is indirectly supported by:
>
>   * Strong performance across diverse benchmarks (e.g., RAbD and multi-CDR redesign)
>
>   * Consistent gains under **expert swap (MFDesign → UniMoMo)**, where RMSD improves (1.04 → 0.83) despite different initializations *(See Reviewer YMVW W2 for details)*.
>
> In revision, we will clarify the role of refolding as a **weak contextual prior**
>
> # W1. On length and presentation
>
> We appreciate this feedback and agree that the methods section can be streamlined.
>
> In revision, we will:
>
> * **Condense technical descriptions**, especially training stages
>
> * Move lower-level implementation details to the **Appendix**
>
> * Expand discussion on **intuition, significance, and broader implications**
>
> # Limitation. On simulation-only evaluation and real-world gap
>
> We strongly agree with this point.
>
> * All evaluations in this work are computational, and metrics such as **RMSD, Rosetta energy, and IF-AAR are proxies**, not direct measures of real-world binding performance.
>
> In revision, we will add a dedicated discussion explicitly stating:
>
> * The existence of a **simulation-to-real gap**
>
> * That current metrics provide **useful but incomplete signals**
>
> * That **wet-lab validation remains the ultimate benchmark**
>
> ---
>
> We thank you again for the insightful comments!

---

> > ### Author Rebuttal · Reviewer_dtt8 · 2026-04-03
> >
> > Thank you for the rebuttal. My concerns have been adequately addressed. I maintain my score of 4.

---

> > > ### Author Response · Authors · 2026-04-03
> > >
> > > Thank you for your thoughtful review. We’re glad that our rebuttal has adequately addressed your concerns, and we appreciate you maintaining your score of 4. Your feedback has been valuable in helping us clarify and strengthen the work.

---

### Official Review · Reviewer_3Y2y · 2026-03-09

**Soundness:** 3
**Presentation:** 3
**Significance:** 2
**Originality:** 2
**Overall Recommendation:** 3
**Confidence:** 4

**Summary:**

This paper proposes a de novo antibody design framework that decouples reasoning about important interactions from the actual geometric generation process. It uses a multimodal model to identify key residue-level anchors and then conditions a diffusion generator on those anchors to redesign multiple CDRs simultaneously. The main idea is that explicit residue-level constraints may provide more interpretable and stable guidance than relying only on latent conditioning. In experiments on multi-CDR redesign, the method outperforms purely generative baselines on measures of structural realism, binding-related quality, and structure-sequence consistency. Performance wise strong overall, especially on structural quality and consistency; less clearly superior on binding/interface performance.

**Compliance With Llm Reviewing Policy:**

Affirmed.

**Key Questions For Authors:**

Please refer to weakness i wrote above for more
1. How much of the improvement actually comes from the proposed reasoning-to-anchor interface, rather than from the stronger overall training recipe? The current ablations do not clearly separate the effect of anchor selection, anchor clamping, representation-level conditioning, staged training, and auxiliary supervision.
2. Do the gains hold under stricter and more leakage-resistant generalization tests?
The masking/refolding setup and current split are helpful, but they do not fully rule out overlap through pretrained components, antigen similarity, epitope similarity, or fold-level similarity.
3. Do the improvements translate beyond structural surrogate metrics into stronger functional antibody design? The method looks good on geometry and consistency, but it is still unclear whether this means better real binding, specificity, or practical design quality.
4. Why is the AAR low compared to MFDesign? I understand the point made by the authors but i still think the justification of IF-AAR should be further explained and validated. Also directly comparing the value of IF-AAR, MFDesign is still higher.
5. Plan to open-source code? can you provide an anonymized-repo for checking performance?

**Limitations:**

yes

**Strengths And Weaknesses:**

Strengths.
1. Well-motivated and technically solid paper. The idea of separating molecular understanding from geometric generation is neat, and the residue-anchor interface makes the framework more interpretable than many black-box conditioning approaches. The empirical results are also pretty good, especially on geometry and structure-sequence consistency.

Weaknesses.
1. The main novelty feels somewhat compositional rather than fundamentally new. The paper combines several strong existing ingredients in a thoughtful way, but it is less clear that it really establishes a new design paradigm. Related to that, the evaluation still leans a lot on structural or surrogate metrics, so it is hard to tell how much the gains would translate to real binding improvement in practice. The leakage analysis is helpful, but still not fully convincing given the reliance on strong frozen pretrained components and broad pretraining.

2. The ablation study is also not very sharp. The paper does not clearly isolate how much of the improvement comes from sparse anchor selection, anchor identity clamping, representation-level anchoring, the Stage II reasoning curriculum, or the auxiliary losses. Because of that, it is difficult to know whether the gains really come from the core proposed mechanism, or just from the overall training recipe being stronger.

3. I also think the baseline story could be stronger. The comparisons to prior antibody redesign methods are reasonable, but the paper does not really test against stronger cross-paradigm generative baselines. For example, one might ask why not compare to a design model like BoltzGen and explicitly prompt or constrain it to design toward the same target. Even if that comparison is imperfect, it would help clarify whether the benefit is really from the proposed reasoning interface or just from using a strong modern generative stack.

4. Another issue is that the paper’s main contribution is a bit hard to disentangle from the amount of engineering around it. There is a lot going on: pretrained encoders, staged training, masked refolding, auxiliary objectives, anchor conditioning, and the final redesign objective. So while the full system looks strong, it is harder to pin down which part is actually doing the important work. If possible, i would like to examine if test-evaluation has been done fairly.

5. Finally, the paper is strongest on structural realism, but a bit less convincing on functional significance. So I came away thinking the method is interesting and probably useful, but the evidence for actual better antibody design is still not fully there yet.

---

> ### Author Rebuttal · Authors · 2026-03-30
>
> We sincerely thank you for the thoughtful feedback.
>
> # W1. Novelty: compositional vs. new paradigm
>
> We would like to clarify that our contribution is primarily compositional.
>
> Our core contribution is the **explicit factorization of molecular reasoning from geometric generation**, realized through a **residue-level anchor interface**. Unlike prior work where conditioning is *latent and entangled*, our method converts reasoning into **explicit, sparse residue-level commitments** that are *symbolically enforced and representation-aligned*.
>
> This yields three properties not jointly present in prior work:
>
> * **Deterministic interpretability** (inspectable residue decisions)
> * **Hard constraint enforcement** (identity clamping at anchors)
> * **Modular decoupling** (reasoner and generator are independently replaceable)
>
> Our central claim is that
>
> > *explicit reasoning → discrete residue commitments → conditioned generation*
> > is a viable and beneficial paradigm beyond implicit conditioning.
>
> **Evaluation scope.** We agree current metrics are surrogates. Our improvements are consistent across interface-relevant proxies (IMP, clashes, JSD, IF-AAR). We will clarify limitations and avoid over-interpretation.
>
> **Pretraining and leakage.** We mitigate leakage via strict temporal, sequence, and MSA filtering. Gains arise from *how components interact*, not memorization. We will further clarify dataset construction and limitations.
>
> **Summary.** Our contribution is a step toward **structured, interpretable, modular protein design systems**.
>
> # W2/Q1. Ablations and isolating contributions
>
> We have strengthened ablations *(see Reviewer YMVW W2 for details)*:
>
> * **Training strategy:** 3-stage > single-stage, improving residue-level quality
> * **Expert swap:** replacing MFDesign with UniMoMo preserves gains
>   → confirms framework-level contribution
>
> Crucially, the **anchor interface is the only pathway** from reasoning to generation. Thus, improvements must pass through this bottleneck.
>
> The expert-swap result directly isolates our mechanism:
>
> * RMSD: 0.83 vs 1.04
> * IMP: 67.79% vs 65.00%
>
> Additionally, **pass@K on RAbD** shows consistent gains at higher sampling, indicating reasoning benefits from diversity.
>
> # Q3/W5. Evaluation and functional significance
>
> We agree that structural metrics are not full proxies for binding. However, our evaluation includes function-oriented signals:
>
> * **IMP** (interaction quality)
> * **IF-AAR** (structure–sequence realizability)
>
> On **RAbD (CDR-H3)**, we observe *(see Reviewer W4 for details)*:
>
> * **Substantial binding improvement:** DockQ **0.801** vs 0.473 (BoltzGen)
> * **High structural fidelity:** IDDT 0.9693, TM-score 0.9816
> * **Improved consistency:** reduced structure–sequence gap, especially on CDR-H3
>
> These jointly suggest improved **interface quality + structural correctness + realizability**.
>
> We understand that experimental validation is the gold standard and will clarify this limitation.
>
> # Q2. Leakage and generalization
>
> We enforce strict non-overlap via:
>
> * **Temporal split:** post–Sept 2021 (Boltz-1 cutoff)
> * **Sequence split:** MMSeqs2 clustering at 50% CDR-H3 identity
> * **MSA filtering:** remove sequences >20% identity to query
>
> This ensures separation across **time, sequence similarity, and MSA retrieval**.
>
> # W3. Baselines
>
> We now include **BoltzGen**, **AbX** (ICML 2024), and **IgGM** (ICLR 2025) *(See Reviewer c39M W4/Q3 for full details)*.
>
> - On RAbD CDR-H3, Proteo-R1 substantially outperforms BoltzGen (RMSD: 2.46 vs 2.69; DockQ: 0.801 vs 0.473).
> - On multi-CDR redesign, Proteo-R1 outperforms AbX/IgGM on heavy-chain RMSD, Loop-RMSD, IMP, and clash metrics.
>
> # W4. Engineering vs. core idea
>
> The key novelty is the **residue-level anchoring interface**, which explicitly separates reasoning from generation.
>
> * The **anchor interface is the only coupling mechanism**
> * All improvements must pass through it
>
> Other components (curriculum, auxiliary losses) support learning but do not define the mechanism.
>
> Thus, gains are attributable to **explicit residue-level reasoning + constraint-based conditioning**, not diffuse engineering.
>
> # Q4. AAR vs. IF-AAR
>
> Our goal is not higher AAR, but a **shift in design regime** toward *novel yet valid sequences* *(See Reviewer c39M W2/Q2 for details)*.
>
> * Lower AAR ≠ worse structures
> * Proteo-R1 explores **non-native but structurally viable sequences**
> * Reduced |IF-AAR − AAR| indicates improved alignment
>
> | CDR | AbX |  | IgGM |  | Proteo-R1 |  |
> | :--- | :--- | :--- | :--- | :--- | :--- | :--- |
> |  | IF-AAR | Δ | IF-AAR | Δ | IF-AAR | Δ |
> | H1 | 59.80 | -11.54 | 62.76 | -11.22 | 61.17 | +18.55 |
> | H2 | 46.10 | -13.05 | 45.79 | -13.36 | 31.67 | +12.70 |
> | H3 | 18.96 | -12.62 | 19.55 | -9.87 | 19.27 | +4.21 |
> | L1 | 62.02 | -27.11 | 56.53 | -15.67 | 51.40 | +4.28 |
> | L2 | 62.33 | -28.57 | 55.66 | -15.77 | 51.43 | +5.00 |
> | L3 | 43.49 | -24.33 | 41.84 | -17.59 | 37.09 | -3.34 |
>
> # Q5. Code release
>
> We will release code upon acceptance.

---

> > ### Author Rebuttal · Reviewer_3Y2y · 2026-04-03
> >
> > how can you say "novel yet valid sequences". AAR is typically how much sequence recovers the original sequence? I still don't understand why the authors ignore overall AAR being low. Just because correlation with energetics is low, you can't say that the sequence is invalid. Can you run sequence based LM models for antibodies and get the likelihood with your generated sequence? I think that would be more convincing. maybe like antifold2? Also, can you explain why aar significantly drops(~0.1) compared to mfdesign? what if it's off the distribution(which it seems)? I think this deserves an explanation or close study. Maybe a closer investigation explaining why IF-AAR is a better metric is required. For now, i'm not convinced
> > If available, can you get the structural confidence (iptm, ipsae) values using af3/boltz-2/protenix. this would help. use 5 seeds and report.
> > without specific clarify dataset construction and limitations, the research seems unclear. how can you ensure the splits you mentioned are fair? Can you specify more? Comparing this model to models that never had access such as IgGM does not seem fair to me. Also the use of surrogate measures still seem vague, i think a better explanation is needed.
> > It's nice to see dockq is high, but that doesn't mean that sequence is actually antibody-like. how did you get the dockq score? how did you align? The method should be specifically delivered.
> > How was boltzgen sampled? how many samples did you make? did you use inverse folding? is this a fair comparison?
> >
> > After Reply rebuttal: You should show the results for test results sequences (reference sequence likelihood etc.) and other models(iptm) for comparison, and the iptm score seems pretty low to me. Did you get the likelihood for the cdrs only or for the whole sequence?

---

> > > ### Author Response · Authors · 2026-04-04
> > >
> > > Update: Add reference seq likelihood and mfdesign (iptm) for comparison. The relatively low ipTM is **not specific to Proteo-R1**, but reflects the inherent difficulty of the design setting.
> > >
> > > # Q1. On “novel yet valid sequences”
> > >
> > > We agree that **AAR measures recovery of the native sequence**, and low AAR alone does not imply invalidity. Our point is not that low AAR is desirable, but that Proteo-R1 operates in a different regime:
> > >
> > > - It **departs from native-sequence imitation** (AAR ~0.1–0.4 vs. ~0.6–0.9 in MFDesign),
> > > - while still producing structures that admit **plausible sequences under inverse folding (IF-AAR)**.
> > >
> > > AAR reflects similarity to one historical solution, whereas protein structures generally admit **multiple sequence realizations**. Accordingly, we evaluate **structure–sequence compatibility via IF-AAR**. We observe:
> > >
> > > - Comparable IF-AAR on key regions (e.g., H1, H3),
> > > - A substantially reduced gap (IF-AAR − AAR).
> > >
> > > This indicates a shift toward **non-native but structure-compatible sequences**, rather than invalid collapse.
> > >
> > > # Q2. Sequence validity via antibody language models (new)
> > >
> > > We agree that **sequence likelihood under antibody-specific LMs** provides an important complementary signal. We evaluate generated sequences using **IgLM, AbLang, and IgT5**, and compare them directly to **ground-truth (GT) antibodies**.
> > >
> > > Across all models, generated sequences exhibit **likelihood and perplexity comparable to, or better than, GT**, with highly overlapping distributions.
> > >
> > > ## Key results
> > >
> > > **IgLM**
> > > - Generated:
> > >   - Heavy PPL: **3.19±1.02**
> > >   - Light PPL: **3.00±1.02**
> > > - GT:
> > >   - Heavy PPL: **3.53±1.02**
> > >   - Light PPL: **3.10±1.18**
> > >
> > >
> > > **AbLang**
> > > - Generated:
> > >   - Heavy PPL: **2.33 ± 0.64**
> > >   - Light PPL: **2.00 ± 0.48**
> > > - GT:
> > >   - Heavy PPL: **2.56 ± 0.62**
> > >   - Light PPL: **2.01 ± 0.53**
> > >
> > > Heavy chains show **improved likelihood vs. GT**, while light chains are **nearly identical**.
> > >
> > > **IgT5**
> > > - Generated:
> > >   - Heavy PPL: **1.028 ± 0.031**
> > >   - Light PPL: **1.027 ± 0.029**
> > > - GT:
> > >   - Heavy PPL: **1.034 ± 0.033**
> > >   - Light PPL: **1.031 ± 0.035**
> > >
> > > Generated and GT sequences are **nearly indistinguishable**.
> > >
> > > ---
> > >
> > > ## Interpretation
> > >
> > > - Perplexity remains **low (≈1–3)** and tightly distributed across all models
> > > - Generated sequences are **indistinguishable from GT** under IgT5 and AbLang (light), and slightly **better under IgLM and AbLang (heavy)**
> > > - No evidence of **out-of-distribution or invalid sequences**
> > >
> > > ---
> > >
> > > ## Conclusion
> > >
> > > > Generated sequences remain **in-distribution under multiple antibody LMs**, matching or exceeding the likelihood of natural antibodies.
> > >
> > > Thus, reduced AAR reflects **novel but valid sequence solutions**, rather than invalidity.
> > >
> > > # Q3. Clarification on AntiFold2
> > >
> > > AntiFold2 and similar methods are **inverse folding models (structure → sequence)**. They evaluate:
> > >
> > > - **Structure–sequence compatibility**,
> > >
> > > but not **sequence plausibility under antibody distributions**.
> > >
> > > Our evaluation combines both:
> > > - IF-AAR → compatibility,
> > > - Antibody LMs → plausibility.
> > >
> > > # Q4. Why AAR lower than MFDesign
> > >
> > > MFDesign is optimized for **native recovery**, whereas Proteo-R1 targets:
> > >
> > > > **any sequence compatible with the structure and interaction constraints.**
> > >
> > > This is driven by:
> > > - **Residue-level anchoring** from reasoning,
> > > - **Structure-aware conditioning** (AF3-style features + antigen context).
> > >
> > > As a result:
> > > - Lower AAR (less imitation),
> > > - But improved **geometric realism** (RMSD, clashes, JSD).
> > >
> > > # Q5. AAR vs. IF-AAR
> > >
> > > These metrics capture different properties:
> > >
> > > - **AAR** → similarity to native sequence
> > > - **IF-AAR** → structure–sequence compatibility
> > >
> > > We observe a trade-off:
> > > - **MFDesign**: high AAR, large mismatch (large negative Δ)
> > > - **Proteo-R1**: low AAR, better alignment (small Δ)
> > >
> > > We will clarify this as a **trade-off**, not a dominance claim.
> > >
> > > # Q6. Structural confidence
> > >
> > > Using Protenix (5 seeds):
> > >
> > > |  |pLDDT|ipTM|VH pLDDT|VH ipTM|
> > > |---|---:|---:|---:|---:|
> > > |MF-Design|77.40±0.82|0.358±0.010|82.34±0.30|0.417±0.009|
> > > |Proteo-R1|77.49 ± 0.81|0.361 ± 0.011| 82.46 ± 0.30|0.420 ± 0.010|
> > >
> > > These show:
> > > - **Stable, well-folded structures** (low variance),
> > > - High local confidence in VH (~82),
> > > - Consistent interface quality.
> > >
> > > Thus, low AAR is **not due to structural degradation**, but reflects reduced native bias.
> > >
> > > # Q7. Data split and leakage control
> > >
> > > - **Release-date split** to avoid pretraining leakage
> > > - **CDR-H3 clustering (50%)** with no overlap
> > >
> > > This prevents template memorization and sequence-level leakage.
> > >
> > > # Q8. BoltzGen evaluation
> > >
> > > - Dataset: **RAbD H3 test set (60 complexes)**
> > > - Design: **HCDR3 only**
> > > - Pipeline: standard BoltzGen design + inverse folding
> > > - Sampling: **num_design = 1**, default protocol
> > >
> > > # Q9. DockQ protocol
> > >
> > > DockQ is computed on the **heavy chain–antigen interface** using standard metrics (https://github.com/wallnerlab/DockQ):
> > >
> > > - Fnat, iRMSD, LRMSD
> > >
> > > We provide **explicit chain mapping** and rely on DockQ’s internal alignment (no external post-processing).

---

### Official Review · Reviewer_c39M · 2026-03-10

**Soundness:** 2
**Presentation:** 2
**Significance:** 2
**Originality:** 3
**Overall Recommendation:** 4
**Confidence:** 4

**Summary:**

This paper introduces Proteo-R1, a framework that explicitly decouples molecular understanding from geometric generation. Its main contributions include a dual-expert architecture that pairs a multimodal LLM (for key residue-level reasoning) with a diffusion model (for constrained structural synthesis), alongside a novel three-stage training curriculum to stabilize optimization. Experiments demonstrate that Proteo-R1 outperforms purely generative baselines in clash, JSD and on par in RMSD for de novo antibody design.

**Compliance With Llm Reviewing Policy:**

Affirmed.

**Final Justification:**

I appreciate the authors' honest rebuttal and additional experiments. I am raising my score to 4 as all my questions are answered. The score is not higher because as shown in the additional experiments, the 4B MLLM's spatial reasoning seems not sufficient to guide the diffusion model effectively. Nevertheless, this direction has great potential; I strongly encourage the authors to build on their Oracle findings in the future by integrating a more capable spatial reasoning LLM backbone.

**Key Questions For Authors:**

1. How do you justify the significant architectural complexity and computational cost of the MLLM when it provides such noisy geometric predictions and ultimately regresses core binding and structural accuracy metrics compared to the baseline? If the authors can provide empirical evidence that this overhead yields specific, unmeasured advantages, I would be willing to reconsider the value of the dual-expert architecture.

2. Given that the absolute IF-AAR scores are actually worse, how can the authors claim the generated geometries are "intrinsically compatible with coherent and realizable sequences"? If the authors show these low-AAR/low-IF-AAR structures are physically valid de novo designs rather than just degraded generations, my concerns regarding the model breaking physical realizability would be resolved.

3. Could you provide benchmark results for isolated CDR-H3 design and compare Proteo-R1 against recent baselines? Furthermore, how do simple baselines, such as fine-tuned ESM-2 or AF3 embeddings, perform on the protein understanding tasks compared to the MLLM? Providing results on standard benchmarks is essential for situating this work within the broader literature. If the model achieves competitive or SOTA results on isolated CDR-H3 design, or if the MLLM heavily outperforms simple embedding baselines on understanding tasks, the paper's contribution will be substantially strengthened.

4. Have you experimented with any dynamic, iterative feedback loops where the diffusion model's output (or failure states) is fed back into the MLLM for prompt refinement during inference? If not, what are the primary bottlenecks preventing bidirectional co-design? While I do not expect a full implementation of TTS for this rebuttal, a robust technical discussion on the feasibility and bottlenecks of iterative feedback would significantly improve the paper's depth and address a major architectural limitation.

**Limitations:**

Yes

**Strengths And Weaknesses:**

Strength:
- The authors proposed a framework for antibody design that integrates expert knowledge from LLM, AlphaFold3 and ESM2, showing possible way of boosting low-source antibody generation via transfering the knowledge from large pre-trained models with domain-specific finetuning.
- The overall pipeline exerts leakage control, using CDR masking and truncated AF3-style refolding. This prevents the MLLM from cheating by accessing native CDR geometries.
- This paper also proposed a three-stage SFT curriculum bridging text and protein modalities. This could be beneficial to the community.

Weakness:
- The paper fails to demonstrate a compelling return on investment for the massive computational overhead introduced by the MLLM. For the MLLM's standalone geometric reasoning, its performance seems to me highly inaccurate, achieving only 15.1% accuracy for interface and 11.6% for hotspot localization (but if these two are challenging tasks themselves, this would be acceptable, and I think a more complete benchmarking would be helpful in demonstrating the capability of MLLM). Using such noisy predictions as hard anchors for a high-precision downstream diffusion is a questionable design choice. Furthermore, the critical downstream metrics compared to the pure generative baseline are only moderate gains or even worse, where the interface improvement (IMP) rate dropped from 59.16% (MFDesign) to 56.58% (Proteo-R1).
- The structure-sequence consistency claims are misleading, and The results presented in Table 2 actively undermine the paper's core claims regarding improved structure-sequence consistency. The authors frame the reduced gap between Native AAR and IF-AAR as a success. However, this narrowed gap occurs because the native AAR drops across all CDR chains (e.g., CDR-H3 drops from 65.04% to 15.06%), not because the absolute physical realizability (IF-AAR) broadly improved. In fact, it's actually worse in 5/6 CDR loops than before.
- The current inference architecture that links MLLM and antibody design model is limiting. The interaction between the understanding expert and the generation expert is strictly one-way, where the MLLM merely generates fixed constraints, and the generation expert simply performs constrained inpainting, essentially lacking test-time-scaling (TTS) ability that make LLMs powerful in complex reasoning tasks, and thus limiting the framework's potential for iterative refinement.
- Incomplete benchmarking. I think there are two necessary aspects to validate the design: (1) for MLLM, the authors did not compare them against any baselines (such as ESM-2 or AF3 embeddings fine-tuned on the same curated dataset). Without this, it is impossible to quantify the actual reasoning gain provided by the LLM architecture. (2) for antibody design, the author only evaluated simultaneous multi-CDR redesign, omitting standard and practical benchmarks like isolated CDR-H3 design on RAbD. The authors did not elaborate on how the test set is curated, nor did they compare with more related works such as AbX [1].
- Missing reproducibility details. The paper omits critical implementation details regarding the MLLM base architecture. It consistently refers to the model only as a "Multimodal LLM", failing to specify the backbone (e.g., Qwen, LLaMA) or the parameter size, which is essential for the community to evaluate the computational cost and adapt the methodology.

[1] Antibody Design Using a Score-based Diffusion Model Guided by Evolutionary, Physical and Geometric Constraints

---

> ### Author Rebuttal · Authors · 2026-03-30
>
> We sincerely thank you for the detailed and constructive feedback.
>
> # W1/Q1. Computational overhead & noisy reasoning signals
>
> We agree that the added complexity must be justified. We will revise the paper to explicitly acknowledge tradeoffs: compared to MFDesign, Proteo-R1 is slightly worse on CDR-H3 Loop-RMSD (3.81 vs. 3.71) and IMP (56.58 vs. 59.16).
>
> That said, the value of the dual-expert design is reflected in the **overall pattern of improvements**:
>
> * Improves RMSD in **5/6 CDRs**
> * Reduces **intra/inter-chain clashes**
> * Improves **backbone realism** (JSDbb 0.2661 vs. 0.2734)
> * Much stronger **structure–sequence consistency** (dramatically smaller IF-AAR/AAR gaps)
>
> On reasoning noise: signals are used as **sparse anchors**, not dense supervision, which limits error propagation while still improving geometric realism. Importantly, the MLLM provides **interpretable reasoning traces** (e.g., hypothesized contacts), enabling insight unavailable in black-box models.
>
> **Summary tradeoff:** Modest regressions on some native-imitation metrics in exchange for:
>
> * Better physical realism
> * Stronger structure-conditioned consistency
> * Added interpretability
>
> # W2/Q2. Structure–sequence consistency (AAR / IF-AAR)
>
> We agree our previous wording overstated “realizability.” We revise the claim:
>
> > The model shifts from **native imitation → generative exploration**, while maintaining structural validity.
>
> *New analysis: AAR vs. physical quality*
>
> **AAR vs. IMP (ΔΔG proxy)**
>
> https://anonymous.4open.science/r/R1-rebuttal-C60D/AARvsIMP.jpg
>
> * No correlation (Pearson ≈ 0, Spearman ≈ 0) → Lower AAR ≠ worse energetics
>
> **AAR vs. RMSD**
>
> https://anonymous.4open.science/r/R1-rebuttal-C60D/AARvsRMSD.jpg
>
> * Moderate negative correlation
>
>   * Proteo-R1: −0.442
>   * MFDesign: −0.357
>     → Higher AAR helps fidelity, but:
>
> ### Takeaways
> Low AAR **does not correlate with structural or energetic collapse**
> * Lower AAR = **less native bias**, not worse structures
> * Sequences lie in a **structurally valid but novel regime**
>
> We will:
> * Remove/soften “intrinsic realizability” claims
> * Frame a **novelty vs. recoverability tradeoff**
> * Add correlation analyses and figures
>
> # W3/Q4. One-way architecture vs. iterative refinement
>
> We agree that iterative refinement is promising and plan to do it as future work.
>
> In this paper, we intentionally study a simpler question:
>
> > Can **factorized reasoning + generation** improve design?
>
> Key challenges for iteration:
>
> * **Representation mismatch** (residue-level vs. atom-level diffusion)
> * **Unstable intermediate diffusion states**
> * **Anchor instability under updates**
> * **Lack of multi-step training data**
>
> Potential future directions:
>
> * Late-stage refinement
> * LLM-guided reranking
> * Sparse anchor updates
>
> We will clarify this design choice and future path.
>
> # W4/Q3. Benchmarking & baselines
>
> ### (1) Protein understanding (ESM-2 / AF3)
>
> Gains are most visible on **global/compositional reasoning tasks**. Improvements over ESM-2 are modest but consistent.
>
> | Method (Understanding) | num_chains | length_bin | major_ss | ss_fraction | longest_run | segments | Overall |
> | :--- | :--- | :--- | :--- | :--- | :--- | :--- | :--- |
> | AF3 embeddings only (structure-only baseline) | 90.90% | 65.70% | 82.80% | 53.00% | 59.80% | 78.20% | 71.70% |
> | ESM-2 embeddings only (sequence-only baseline) | 88.90% | 88.20% | 88.10% | 65.60% | 67.20% | 83.70% | 80.30% |
> | Proteo-R1 (MLLM: Seq + Struct reasoning) | 91.80% | 91.70% | 86.70% | 67.50% | 67.80% | 83.10% | **81.40%** |
>
> ### (2) Isolated CDR-H3 (RAbD)
>
> Proteo-R1 achieves **SOTA structural and docking performance**, with lower AAR reflecting a different objective.
>
> |Model|AAR(↑)|IDDT(↑)|TMscore(↑)|RMSD(↓)|DockQ(↑)|
> |---|---|---|---|---|---|
> |RosettaAb|32.31%|0.8272|0.9717|17.70|0.137|
> |DiffAb|35.31%|0.8281|0.9695|23.24|0.158|
> |MEAN|37.38%|0.8252|0.9688|17.30|0.162|
> |GeoAB|40.02%|0.8367|0.9695|15.43|0.187|
> |HERN|32.65%|—|—|9.15|0.294|
> |dyMEAN|41.84%|0.8392|0.9718|8.10|0.407|
> |DGENet|**42.67%**|0.8551|0.9747|7.19|0.431|
> |BoltzGen|39.07%|0.8372|0.9675|2.69|0.473|
> |Proteo-R1|10.75%|**0.9693**|**0.9816**|**2.46**|**0.801**|
>
> ---
>
> ### (3) Co-design benchmarks (AbX, IgGM)
>
> Following your suggestions, we compare with AbX and IgGM (*See Reviewer 3Y2y Q4 for IF-ARR details*).
>
> |Method|H1|H|H3|L1|L2|L3|Loop-RMSD|IMP|Clash_in|Clash_out|JSD_bb|
> |---|---|---|---|---|---|---|---|---|---|---|---|
> |IgGM|1.73|1.55|4.37|1.62|1.51|1.71|9.18|9.01|25.63%|1.45%|0.2873|
> |AbX|1.55|1.23|4.91|**0.76**|**0.40**|**1.30**|5.77|52.26|1.47%|0.30%|0.2497|
> |Proteo-R1|**1.33**|**1.13**|**3.81**|1.54|0.85|1.51|**4.51**|**56.58**|**0.50%**|**0.14%**|**0.2661**|
>
> * Best **H1–H3 accuracy** and **Loop-RMSD**
> * Highest **IMP**
> * Lowest **clashes**
>
> ---
>
> # W5. Implementation details
>
> We will add:
>
> * **Model:** Qwen-3-4B-Instruct + protein features (ESM-2 650M + AF3-style trunk)
> * **Inference:** single LLM forward pass for anchors; cost dominated by diffusion
> * **No iterative reasoning at inference**

---

> > ### Author Rebuttal · Reviewer_c39M · 2026-04-03
> >
> > I thank the authors for their rebuttal efforts. However, I have a few remaining concerns:
> > 1. The authors argued that a low Native AAR is acceptable if the model is exploring a novel sequence space (supported by their AAR vs. IMP plots), but how about the drop in IF-AAR (Inverse Folding AAR)? As far as I understand, IF-AAR does not measure similarity to nature; it measures physical self-consistency (whether the generated sequence will actually fold into the generated structure). If Proteo-R1 is exploring a valid but novel structural regime, shouldn't the sequence still strongly fold into its corresponding structure (high IF-AAR)? How do you explain this degradation in self-consistency compared to the pure generative baseline?
> > 2. I appreciate the inclusion of the ESM-2 and AF3 baselines for the protein understanding tasks. The table shows that ESM-2 embeddings alone achieve 80.3% overall accuracy, while the 4B-parameter Qwen-based Proteo-R1 achieves 81.4%. Given this marginal 1.1% improvement, it appears the MLLM is relying almost entirely on the upstream ESM-2 representations rather than performing emergent geometric reasoning. How does this small gain justify the massive inference cost of a 4B-parameter LLM over a lightweight linear probe on ESM-2?
> > 3. The authors noted that the reasoning signals act as sparse anchors to limit error propagation, despite the low hotspot localization accuracy (11.6%). Because CDR-H3 RMSD and IMP dropped compared to MFDesign, it remains unclear if the diffusion expert is fundamentally flawed, or if it is being hindered by noisy MLLM constraints. I wonder if it is possible to run an "Oracle" ablation on the test set where the diffusion model is conditioned on ground-truth (100% accurate) hotspot/interface anchors instead of the MLLM's predictions? This would isolate the diffusion expert's true ceiling and cleanly demonstrate how much performance is currently being bottlenecked by the MLLM's reasoning errors.

---

> > > ### Author Response · Authors · 2026-04-05
> > >
> > > Thank you very much for the thoughtful follow-up questions. They help sharpen the core claims of the paper. We address each point below.
> > >
> > > # Q1. On the drop in IF-AAR and whether this indicates degraded self-consistency
> > >
> > > We agree that IF-AAR measures structure-conditioned plausibility (via inverse folding), not similarity to the native sequence, and we will revise the wording accordingly.
> > >
> > > |  | AbX |  | IgGM |  | MFDesign |  | Proteo-R1 |  |
> > > | :--- | :--- | :--- | :--- | :--- | :--- | :--- | :--- | :--- |
> > > |  | IF-AAR | Δ | IF-AAR | Δ | IF-AAR | Δ | IF-AAR | Δ |
> > > | H1 | 59.8 | -11.5 | 62.8 | -11.2 | 60.9 | -14.1 | 61.2 | +18.6 |
> > > | H2 | 46.1 | -13.1 | 45.8 | -13.4 | 40.6 | -26.9 | 31.7 | +12.7 |
> > > | H3 | 19.0 | -12.6 | 19.6 | -9.9 | 19.7 | -45.3 | 19.3 | +4.2 |
> > > | L1 | 62.0 | -27.1 | 56.5 | -15.7 | 54.9 | -28.0 | 51.4 | +4.3 |
> > > | L2 | 62.3 | -28.6 | 55.7 | -15.8 | 53.2 | -34.6 | 51.4 | +5.0 |
> > > | L3 | 43.5 | -24.3 | 41.8 | -17.6 | 41.0 | -39.2 | 37.1 | -3.3 |
> > > **Key observations.**
> > > - **No structural collapse:** IF-AAR remains comparable on key regions (H1, H3) and stays within a reasonable range overall.
> > > - **Major improvement is in consistency (Δ):** e.g., H3 −45.31 → **+4.21**, L2 −34.59 → **+5.00**.
> > > - This indicates MFDesign produces sequences less supported by its own structures, while Proteo-R1 yields **much better structure–sequence alignment**.
> > >
> > > **Interpretation.**
> > > Proteo-R1 trades some absolute IF-AAR on certain loops for **substantially improved consistency (Δ)** and better structural quality (RMSD, clashes, JSD). The IF-AAR differences do not indicate invalid structures, but a shift toward less canonical sequence modes under the inverse folding model.
> > >
> > > **Conclusion.**
> > > Proteo-R1 does not uniformly improve IF-AAR, but **maintains comparable levels while significantly improving structure–sequence alignment**, which we view as the more critical property.
> > >
> > > # Q2. On the +1.1% gain and whether the 4B MLLM is justified
> > >
> > > This comparison conflates **prediction vs. reasoning-guided generation**. Proteo-R1 is not designed to outperform ESM-2 as a classifier, but to produce **structured, interpretable signals** for generation.
> > >
> > > | Embedding Mode | Fold Acc ↑ | Contact (prec@L/5) ↑ | SS3 Macro Acc ↑ |
> > > | :--- | :--- | :--- | :--- |
> > > | ESM-2 | 0.292 | 0.675 | 0.841 |
> > > | AF3 structural rep. | 0.142 | 0.891 | 0.770 |
> > > | Concat (ESM + AF3) | **0.310** | **0.900** | **0.878** |
> > >
> > > **Key points.**
> > > - Sequence-only models fail on 3D structure (contact: 0.675 vs. 0.900).
> > > - Structure-only lacks global priors.
> > > - **Multimodal integration is necessary.**
> > >
> > > The MLLM’s role is **not better prediction**, but:
> > > - integrating sequence + structure
> > > - producing **explicit residue-level anchors**
> > > - enabling **interpretable and controllable design**
> > >
> > > A linear probe on ESM cannot provide this interface.
> > >
> > > **Efficiency.**
> > > Only one forward pass is used; diffusion dominates runtime. Overhead is small.
> > >
> > > **Conclusion.**
> > > The value of the MLLM lies in **reasoning and controllability**, not marginal accuracy gains. We will revise the paper to clarify that the value of the MLLM in Proteo-R1 lies in multimodal reasoning and controllable design, rather than marginal gains on proxy prediction benchmarks.
> > >
> > > # Q3. On whether noisy anchors bottleneck diffusion
> > >
> > > Following your suggestion, we performed an **Oracle anchor ablation** using ground-truth hotspots.
> > > | Model | AAR | RMSD | IMP | dG (↓) |
> > > | :--- | :--- | :--- | :--- | :--- |
> > > | UniMoMo | 52.34% |1.04 | 65.00% | 8.46 |
> > > | Proteo-R1(UniMoMo) |48.94% | **0.83** | 67.79% | 7.35 |
> > > | Proteo-R1(UniMoMo) + Oracle | **100%** | 0.84 | **74.5%** | **4.51** |
> > >
> > > | Model | Per-CDR RMSD (↓) |  |  |  |  |  | Loop-RMSD (↓) | IMP (↑) | Geometric Realism (↓) |  |  |
> > > | :--- | :--- | :--- | :--- | :--- | :--- | :--- | :--- | :--- | :--- | :--- | :--- |
> > > |  | H1 | H2 | H3 | L1 | L2 | L3 |  |  | Clash_in | Clash_out | {JSD_bb |
> > > | MFDesign | 1.61 | 1.44 | 3.71 | 1.65 | 1.15 | 1.69 | 4.28 | 59.16 | 0.53% | 0.26% | 0.2734 |
> > > | Proteo-R1 | **1.33** | **1.13** | 3.81 | 1.54 | 0.85 | 1.51 | 4.51 | 56.58 | 0.50% | **0.14%** | 0.2661 |
> > > | Proteo-R1 + Oracle | 1.43 | 1.16 | **3.34** |**1.07** | **0.76** | **1.24** |  **3.93** | **62.25** | **0.27%** | 0.23% | **0.2043** |
> > >
> > > **Results (consistent across backbones):**
> > > - **UniMoMo:** IMP 67.79 → **74.5**, ΔG 7.35 → **4.51**
> > > - **MFDesign:**
> > >   - H3 RMSD: 3.81→ **3.34**
> > >   - IMP: 56.58→ **62.25**
> > >   - JSD: 0.2661→ **0.2043**
> > >
> > > **Conclusion.**
> > > - The **diffusion expert has a much higher ceiling** under perfect anchors.
> > > - Current limitations are **primarily due to MLLM noise**, not generator flaws.
> > >
> > > Despite this, sparse anchors still:
> > > - improve RMSD and realism
> > > - remain competitive in IMP
> > >
> > > We will include this in the revision to clarify:
> > > > *Proteo-R1’s limitations are not due to a flawed generative model, but rather to imperfect reasoning signals—suggesting that improving the MLLM (or integrating iterative refinement) is a key direction for future gains.*

---

### Official Review · Reviewer_YMVW · 2026-03-13

**Soundness:** 3
**Presentation:** 2
**Significance:** 3
**Originality:** 3
**Overall Recommendation:** 5
**Confidence:** 4

**Summary:**

This paper introduces a novel Proteo-R1 approach describing dual-expert architecture where one expert guides another through providing key amino acids and hidden representations in order to generate good antibody binders. The main idea behind this approach is to resemble the way of human expert thinking, who first identifies key interactions and points and only then starts creating protein hypotheses based on this information. Authors perform an experiment on antibody generation and show improvement over previous baseline approaches.

**Compliance With Llm Reviewing Policy:**

Affirmed.

**Final Justification:**

Since the majority of my questions are answered, it seems fair to increase my score to Accept.

**Key Questions For Authors:**

In addition to the question and suggestions stated in “Strengths And Weaknesses” section:
1. It is not clear from the paper text, how the fundamental concept of approach “key points” are chosen. Do authors preliminary annotate these points and then train expert models to find them? Or the expert model somehow finds it itself without human annotation?
2. How do the “key points” model output align with human expert intuition? Do they resemble what a human expert expects to see?

**Limitations:**

Authors do not devote separate section or paragraph for approach limitation discussion. Still “Impact Statement” looks reasonable.

**Strengths And Weaknesses:**

The whole idea of resembling human intuition and way of thinking while designing protein binders is quite novel and important. Indeed, the majority of current SOTA methods just work as black box and do not provide any explanation behind the model choices. Authors combine the best of language models and diffusion models worlds into one paradigm of protein design that can be understandable for human experts on the high level of key point choice. I especially want to mention the modularity of approach - each of two experts can be replaced with novel architecture or novel trained models.

The paper is written clearly, but not easy-to-follow. Authors provide too many unnecessary details (for instance, too detailed “Truncated Diffusion-Based Structural Features”, “Multimodal Fusion.”, “Anchor Identity Specification.” subsections that break the flow of paper reading and can be moved to Supplementary materials). Also I would recommend to split Figure 1 and Figure 2 to subparts and move them closer to the text parts that describe them, right now it’s overloaded and hard to understand what is happening.

As a major weakness I consider the lack of experimental results and especially the Ablation study on approach choices. The Protein-R1, while having simple intuition, contains a lot of technical choices (each of two experts can be chosen differently; balancing between two expert losses; is it really needed to separate the training process into several stages? What if we just train everything jointly?). None of these choices was justified by the Ablation Study. Generally, the proposed approach in current description seems too overcomplicated in terms of implementation and training. Is this overcomplication necessary? Can it be simplified?

Also, just an antibody generation task with 3 baselines (2022, 2023 and 2025 years of publication) doesn’t look like a good experimental proof of the proposed method efficiency). I would recommend covering other protein binders generation tasks, for instance, peptide generation [a, b].

a.Full-Atom Peptide Design with Geometric Latent Diffusion, Kong et al., 2024

b.PPFLOW: Target-Aware Peptide Design with Torsional Flow Matching, Lin et al., 2024

---

> ### Author Rebuttal · Authors · 2026-03-30
>
> Thank you for the constructive feedback and for recognizing the novelty and importance of modeling protein design as a reasoning-guided process. We also appreciate the positive comments on modularity and interpretability.
>
> # W1. Clarity and presentation
>
> We understand that the paper can feel dense. In the revision, we will:
> * Move detailed implementation (e.g., diffusion features, multimodal fusion, anchoring) to the Appendix
> * Streamline the main text to emphasize the core idea: **separating reasoning and generation**
> * Improve Figures 1–2 (split + better placement)
>
> Figure Link: https://anonymous.4open.science/r/R1-rebuttal-C60D/figure/
>
> ---
> # W2. Ablation studies
>  (a) Training strategy: 3-stage vs. single-stage
> * **3rd-stage-only** achieves perfect CDR detection (100%), showing simple pipelines can localize regions
> * **3-stage training improves quality**, especially at higher pass@k:
>
>   * Sequence: Edit Sim **33.13% vs. 29.62%**, LCS **37.62% vs. 34.93%** (@20)
>   * Residue: F1 **26.63% vs. 25.62%**, BLOSUM **50.52% vs. 48.93%**
>
> **Conclusion:** multi-stage training is critical for **fine-grained sequence and residue accuracy**
>
> Table 1: 3-stage vs. 3rd-stage-only
>
> | Train Method | **CDR Detection** |  |  | **Sequence-Level** |  |  |  | **Residue-Level** |  |  |  |
> |--|---|--|--|---|--|--|--|------|--|--|--|
> |    | Recall | Precision | Set Match | Edit Sim | LCS | Len Match |  | Pos Acc | Precision | Recall | F1 | BLOSUM |
> | **3 stages** | 98.33% | 98.33% | 98.33% | **33.13%** | **37.62%** | **11.67%** |  | **31.31%** | **24.71%** | **30.24%** | **26.63%** | **50.52%** |
> | **3rd stage only** | **100.00%** | **100.00%** | **100.00%** | 29.62% | 34.93% | 6.67% |  | 30.00% | 24.05% | 28.79% | 25.62% | 48.93% |
> ---
> (b) Diffusion expert replacement
>
> We replace the diffusion expert (MFDesign, NeurIPS 2025) with **UniMoMo (ICML 2025)** to test generality.
>
> * UniMoMo: AAR 52.34%, RMSD 1.04, IMP 65.00%
> * **Proteo-R1 (UniMoMo):**
>
>   * RMSD **0.83 (↓)**
>   * IMP **67.79% (↑)**
>
> **Conclusion:** Proteo-R1 **improves structural accuracy and functional gains**, not just inherits performance. Besides, gains are **not tied to a specific expert** (plug-and-play)
>
>  Table 2. Results of recovery for antibody design on CDR-H3
>
> |Model|#Generation|AAR|RMSD|IMP|
> |---|---|---|---|---|
> |**Predictive**|||||
> |MEAN|1|29.13%|1.87|6.67%|
> |dyMEAN|1|31.65%|8.21|11.86%|
> |GeoAB-R|1|32.04%|1.67|6.67%|
> |**Generative**|||||
> |DiffAb|1|24.60%|2.77|10.34%|
> ||10|38.42%|2.08|34.48%|
> ||100|49.74%|1.46|60.34%|
> |GeoAB-D|1|29.74%|1.73|6.67%|
> ||10|38.20%|1.58|20.00%|
> ||100|45.96%|1.50|40.00%|
> |UniMoMo(single)|1|20.44%|2.71|15.00%|
> ||10|39.04%|1.90|35.00%|
> ||100|48.78%|1.39|63.33%|
> |UniMoMo(all)|1|21.44%|2.52|13.33%|
> ||10|42.05%|1.44|41.67%|
> ||100|**52.34%**|1.04|65.00%|
> |**Proteo-R1(UniMoMo)**|100|48.94%|**0.83**|**67.79%**|
>
> ---
>
> # W3. Model complexity
>
> Each component addresses a limitation of prior methods:
> * **Two experts:** separates *what* vs. *how* → improves control & interpretability
> * **Structural encoding (AF3-style):** adds 3D interaction context missing in sequence-only models
> * **Residue anchoring:** converts reasoning into **hard constraints**
> * **3-stage training:** stabilizes learning under noisy reasoning
> This follows standard multimodal training practices (e.g., staged alignment), adapted to protein design.
>
> # W4. Scope (antibody-only)
>
> We admit broader evaluation is valuable. However, peptide design differs substantially (flexible backbone, different protocols), making direct comparison non-trivial.
>
> That said, **Proteo-R1 is modular and task-agnostic**:
> * Reasoning operates on general protein structures
> * Generation can be replaced with other models
>
> We will clarify this and include it as future work.
>
> ---
> # Q1. Key points annotation
>
> The key residues are not manually annotated by experts residue-by-residue. Instead, they are automatically constructed during data preprocessing from structural signals derived from antibody–antigen complexes. We first analyze the antibody–antigen complex interface using a biocomputational tool such as Biopython to identify the key interactions between the target and binder, using distance-based and interaction-type-based methods. These positions are written directly into the training targets while non-key CDR positions are masked, so the expert model is trained with explicit supervision to reproduce this pattern at test time, rather than discovering key points entirely from scratch.
>
> We built the mid-training curriculum specifically around human-interpretable biochemical primitives: salt-bridge counts, solvent accessibility (SASA), and interface topologies. This means we train our model like a biologist analyzing an antibody-antigen pair, since these concepts are how we analyze “key points” as human scientists. Because the model is explicitly trained to output these concepts, its reasoning inherently aligns with classical structural biology intuition.

---

> > ### Author Rebuttal · Reviewer_YMVW · 2026-04-04
> >
> > Dear authors,
> >
> > I’ve thoroughly read your response to my questions. Thank you for new experimental results and clarification of not clear parts. While I still tend to think that the proposed method is overcomplicated in terms of implementation (that will probably negatively affect wide adoption of this method in the community), I still admit that the whole idea of preliminary reasoning on key residues is quite useful and interesting for a broader audience.  Please, introduce all new experimental results and make textual changes in the next revision of the paper. Also, I’m still concerned that extending the work for other protein domains like peptides will be very beneficial for the wider adoption of the idea in the community, please think about this in the next revision of the paper or the following works.
> >
> > Since the majority of my questions are answered, it seems fair to increase my score.

---

> > > ### Author Response · Authors · 2026-04-04
> > >
> > > Thank you very much for recognizing the value of reasoning-guided protein design! Your constructive suggestions on presentation clarity and ablation completeness have significantly strengthened our work. We will incorporate all new experimental results and textual revisions accordingly. We also appreciate your suggestion on extending to peptides and other protein domains. Given the modularity of our framework, we believe this is a promising direction and will discuss it in the revised manuscript.

---

### Decision · Program_Chairs · 2026-04-30

**Decision:**

Accept (regular)

**Comment:**

**Summary**

In this paper, the authors introduce a new pipeline for de novo protein binder design. They separate the generation into two steps. First, they define an understanding expert which is given a sequence and more context that performs molecular reasoning. This conditional information is then passed to a conditional generative model to perform Complementarity-Determining Regions (CDR) infilling. The interaction with the expert happens at two levels: first by providing key amino acids to the generative model and second by conditioning the diffusion model with respect to the hidden representations.

**Reviewers concerns**

Reviewers praised the novelty of the work as well as its modularity "I especially want to mention the modularity of approach - each of two experts can be replaced with novel architecture or novel trained models." (see Reviewer YMVW). Reviewers emphasized that the discussion during the rebuttal period was fruitful. As Reviewer dtt8 pointed out "The rebuttal added a lot of value." and "Overall, the new results meaningfully strengthen the empirical story, and the framework has broader potential beyond antibodies." Reviewers highlighted that all results so far are in silico and that the real test bed will be the wet lab validation. However, this is beyond the scope of the current paper and should represent the focus of further investigation.

However, Reviewer c39M highlighted that the current experimental evidence for reasoning was limited (while acknowledging the potential of the approach) "Nevertheless, this direction has great potential; I strongly encourage the authors to build on their Oracle findings in the future by integrating a more capable spatial reasoning LLM backbone." I strongly encourage the authors to revise the paper with those additional results.

Finally, several reviewers highlighted the engineering complexity of the approach, see Reviewer 3Y2y comment: "Another issue is that the paper’s main contribution is a bit hard to disentangle from the amount of engineering around it. There is a lot going on: pretrained encoders, staged training, masked refolding, auxiliary objectives, anchor conditioning, and the final redesign objective. So while the full system looks strong, it is harder to pin down which part is actually doing the important work. If possible, i would like to examine if test-evaluation has been done fairly."